# C-band radar data and in situ measurements for the monitoring of wheat crops in a semi-arid area (center of Morocco)

Nadia Ouaadi[1,2], Jamal Ezzahar[3,4], Saïd Khabba[1,4], Salah Er-Raki[4,5], Adnane Chakir[1], Bouchra Ait Hssaine[4], Valérie Le Dantec[3], Zoubair Rafi[1,2], Antoine Beaumont[6], Mohamed Kasbani[3], Lionel Jarlan[3]

[1]LMFE, Department of Physics, Faculty of Sciences Semlalia, Cadi Ayyad University, Marrakech, Morocco

[2]CESBIO, University of Toulouse, IRD/CNRS/UPS/CNES, Toulouse, France

[3]MISCOM, National School of Applied Sciences, Cadi Ayyad University, Safi, Morocco.

[4]CRSA, Mohammed VI Polytechnic University UM6P, Benguerir, Morocco.

[5]ProcEDE, Department of Applied Physics, Faculty of Sciences and Technologies, Cadi Ayyad University, Marrakech, Morocco.

[6]Atmo Hauts-de-France, Lille, France.

*Correspondence to*: J. Ezzahar (j.ezzahar@uca.ma)

**Abstract.** A better understanding of the hydrological functioning of irrigated crops using remote sensing observations is of prime importance in the semi-arid areas where the water resources are limited. Radar observations, available at high resolution and revisit time since the launch of Sentinel-1 in 2014, have shown great potential for the monitoring of the water content of the upper soil and of the canopy. In this paper, a complete set of data for radar signal analysis is shared to the scientific community for the first time to our knowledge. The data set is composed of Sentinel-1 products and *in situ* measurements of soil and vegetation variables collected during three agricultural seasons over drip-irrigated winter wheat in the Haouz plain in Morocco. The *in situ* data gathers soil measurements (time series of half-hourly surface soil moisture, surface roughness and agricultural practices) and vegetation measurements collected every week/two weeks including above-ground fresh and dry biomasses, vegetation water content based on destructive measurements, cover fraction, leaf area index and plant height. Radar data are the backscattering coefficient and the interferometric coherence derived from Sentinel-1 GRDH (Ground Range Detected High resolution) and SLC (Single Look Complex) products, respectively. The normalized difference vegetation index derived from Sentinel-2 data based on Level-2A (surface reflectance and cloud mask) atmospheric effects-corrected products is also provided. This database, which is the first of its kind made available in open access, is described here comprehensively in order to help the scientific community to evaluate and to develop new or existing remote sensing algorithms for monitoring wheat canopy under semi-arid conditions. The data set is particularly relevant for the development of radar applications including surface soil moisture and vegetation parameters retrieval using either physically based or empirical approaches such as machine and deep learning algorithms.

The database is archived in the DataSuds repository and is freely-accessible via the DOI: https://doi.org/10.23708/8D6WQC (Ouaadi et al., 2020a).

## 1 Introduction

The south-Mediterranean region has been identified as a hot spot of climate change (Giorgi, 2006; Giorgi and Lionello, 2008; IPCC, 2014) that may worsen the water shortage already affecting the region. Up to 90% of available water is dedicated to irrigation (Ministre de l'agriculture et peche maritime du develpement rurale et des eaux et forets, 2018). Indeed, the predicted temperature rise that could reach 3°C by 2050 combined to precipitation decrease and increased evapotranspiration could drastically increase the irrigation requirements. The demand for water is also already increasing in response to an ever-growing population and to changes of agricultural practices-intensification, conversion to cash crops, rise of irrigated areas (Ducrot et al., 2004; Fader et al., 2016; Jarlan et al., 2016). The monitoring of irrigated crops and the optimization of water use is therefore of prime importance for the sustainability of the water resources in the Mediterranean region. This requires the implementation of methods to monitor the crop water status and the underlying soil moisture (Wang et al., 2012).

Within this context, the observations from active spaceborne sensors in the microwave domain (radar) have shown great potential for the monitoring of crops (Mattia et al., 2003; Ouaadi et al., 2020b; Picard et al., 2003). The potential of radar data for monitoring irrigated crops originates from their high sensitivity to the water status of the surface including the water content of the above ground biomass and the moisture of the upper soil layer (Ulaby and Dobson, 1986). It is also sensitive to the structural properties of the observed target including the size and orientation of the canopy elements (leaves, steams, trunks) and the soil roughness. A key advantage of radar observations for monitoring crops, especially those crops growing during the rainy season such as wheat, is also that it is not prone to atmospheric perturbations. Sentinel-1 provides for the first time since 2014 backscattering coefficients at a resolution of 10 m and a revisit time of 6 days compatible with the high dynamic of annual crops at the field scale paving the way to an operational use of C-band radar data for crop monitoring.

Nevertheless, radar signal is a complex mix of backscattering from the soil and from the canopy that are often difficult to disentangle. The impact of any changes of the canopy structure such as the appearance of the heads during the heading stage of wheat (Brown et al., 2003; El Hajj et al., 2019; Ulaby et al., 1986) or of the soil roughness may also drastically impact the backscattering response. These processes are not fully understood and not always properly reproduced by the backscattering models.

The sensitivity of the backscattering coefficient to the surface soil moisture (SSM) is widely documented in the literature for bare or covered soils (Ezzahar et al., 2020; Ouaadi et al., 2020c, 2020b; Ulaby and Dobson, 1986; Zribi et al., 2014). Several retrieval approaches based on the inversion of a radiative transfer models (Bai et al., 2017; Gherboudj et al., 2011; El Hajj et al., 2016; Li and Wang, 2018; Ouaadi et al., 2020b) or based on linear or non-linear empirical regression (Gorrab et al., 2015; Ouaadi et al., 2020b) have been developed. The SSM derived from radar observations are also used to estimate RZSM (root zone soil moisture), a key variable in agronomy, through the combination with a land surface model (Cho et al., 2015; Das et al., 2008; Dumedah et al., 2015; Ford et al., 2014; Rodell et al., 2004; Sabater et al., 2006; Sure and Dikshit, 2019).

The presence of a canopy above the soil results in two more contributions to the backscattered signal: the volume scattering
and the attenuated signal by the canopy. The water content of vegetation influences the dielectric properties, that in turn
influence the radar backscatter from the vegetation (Ulaby et al., 1982). Based on these findings, some studies are focused on
the retrieval of vegetation parameters from SAR (Synthetic Aperture Radar) data such as above-ground biomass (Hosseini
and McNairn, 2017; Periasamy, 2018; Taconet et al., 1994) or even grain yield (Fieuzal et al., 2013; Patel et al., 2006). In
addition to the backscattering coefficient, the polarization ratio and the interferometric coherence have demonstrated
potentialities for the characterization of the vegetation including height (Blaes and Defourny, 2003; Engdahl et al., 2001),
vegetation cover fraction (Wegmuller and Werner, 1997), fresh above-ground biomass (Mattia et al., 2003; Veloso et al.,
2017), above-ground biomass (Ouaadi et al., 2020b) and vegetation water content (Ouaadi et al., 2020b). Other studies
acknowledge the sensitivity of coherence to soil moisture (De Zan et al., 2014; Scott et al., 2017). Recent research suggests
that radar observations could also provide valuable information on the canopy water status (Van Emmerik et al., 2015;
Ouaadi et al., 2020d) for crop stress detection.
In situ measurements of vegetation and soil characteristics are always needed to improve our understanding of the radar
response, to develop and calibrate radiative transfer models and to propose generic retrieval methods for the inversion of soil
or vegetation variables. Nevertheless, in situ data set dedicated to these objectives are really specific in the sense that, for
instance, soil roughness is only of interest for understanding the physical principle of observations in the microwave domain.
Likewise, above-ground biomass is often measured by agronomist for crop modeling for instance but the partition between
dry and wet matter, a key variable for radar acquisition, is hardly ever done. Indeed, the latter relies on heavy destructive
measurements consisting in cutting all the vegetation elements within squares sample in the field and a double weighting
before and after drying the samples in an oven. In this paper, a recent, multi-year and complete database composed of
processed Sentinel-1 SAR data (the backscattering coefficient and the interferometric coherence), Sentinel-2 NDVI and
measured variables on the soil, on the vegetation and on the agricultural practices are made available. The in situ data
include automatic measurements as well as observations carried out during measurement campaigns once or twice every 15
days throughout the growing season. This database covers 3 wheat seasons (2016-2017 to 2018-2019) of 3 different irrigated
fields (Ouaadi et al., 2020b). It is a unique and valuable data set that can be used for vegetation and soil moisture monitoring
applications including from radar observations. In addition, the multiyear database can be useful for multiyear time series
analysis. In the next section, an overview of the field-location and a detailed description of the variables, including field
measurements and remote sensing data processing, are presented. In Section 3, the variables are experimentally and
physically analyzed to assess the consistency of the dataset. Conclusions are provided in Section 4.

## 2 Study area and experimental sites

### 2.1 Study area

The database described in this paper is collected in the Haouz plain in the Tensift watershed, center of Morocco (Fig. 1). This plain is one of the most important plains in Morocco located at 550 m above sea level and covers about 6000 km$^2$ of which 2000 km$^2$ are irrigated. The climate in the region is Mediterranean semi-arid, with an annual average precipitation of about 250 mm. The distribution of precipitations highlights a wet season with around 85% of annual precipitations between October and April, and a dry season from May to September. The maximum average of temperature occurs during summer in July-August (about 35°C) and the minimum in January (about 5°C) (Abourida et al., 2008). The average air humidity is about 50% and the reference evapotranspiration ET0 is around 1600 mm/year (Jarlan et al., 2015), which is greatly exceeding the annual rainfall. The agricultural production in the plain is not very diverse, focusing on cereals (51% of the irrigated areas), olive trees (30% of the irrigated area), 9% of fodder production and 2% of market gardening for cattle breeding while the non-irrigated part of the plain is cropped with rainfed wheat (Abourida et al., 2008). Wheat is usually sown between November and January depending on precipitation distribution, even for irrigated field, and on cultivar. Harvest usually occurs in May or June.

### 2.2 Experimental sites

The database concerns three irrigated fields (F1, F2 and F3) located within a private farm in the province of Chichaoua located 65 Km west of Marrakech city (Fig. 1). F1 and F2 are monitored during two successive growing seasons (2016-2017 and 2017-2018) while F3 is monitored during the season 2018-2019. The fields are sown using an automatic seed drill. They are irrigated using the drip technique. For all the fields, the wheat is cropped once a year during winter-spring (see Table 1 for sowing and harvest dates). After harvest, the fields are generally used for cattle grazing until mid-July when the plowing works started. Table 1 summarizes some general information about the fields. Please note that during the 2017-2018 season, wheat in F2 is affected by specific growing conditions: i) the development of adventices belonging to the wild thistles family characterized by a horizontal structure, ii) the seeding density is higher than in F1, and iii) the seeding is a mixture of barley and wheat within F2. This resulted in very long stems: 146 cm in F2 compared to 110 cm in F1 in April 2018. Finally, these long stems in F2 are laid down by the wind from April 12, 2018. A picture of F2 during 2017-2018 is provided in appendix A (Fig. A1). Although such exceptional growing conditions are not very likely, it has been chosen to include this crop season in the data set to cover different conditions of growth.


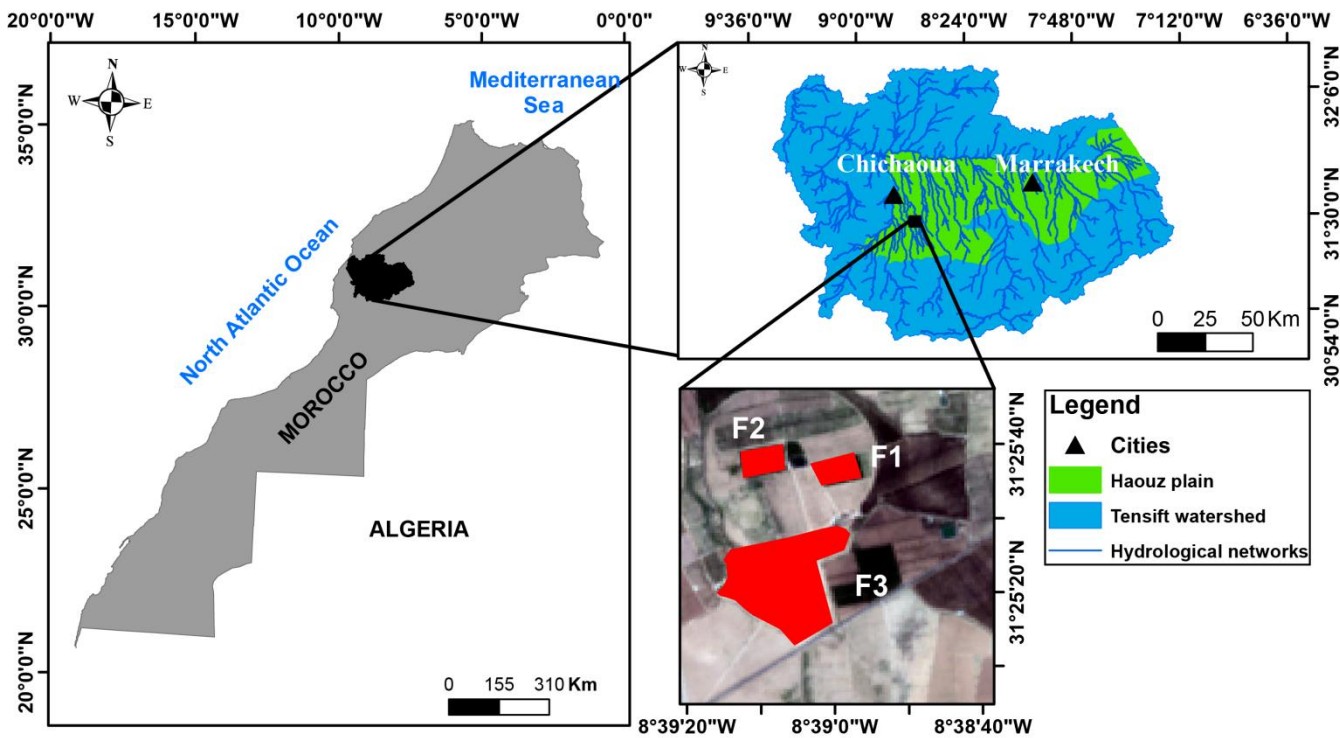

**Figure 1. Location of the study fields: F1, F2 and F3 are drip-irrigated wheat plots in a private farm ("Domaine Rafi") near**
**Chichaoua city in the Haouz plain, center of Morocco.**
**Table 1. General information about the three fields**

| Field | Area (ha) | Season | Sowing date | Harvest date | Irrigation | Sand (%) | Clay(%) |
|-------|-----------|--------|-------------|--------------|------------|----------|---------|
| **F1** | 1.5 | 2016-2017 & 2017-2018 | Nov 25, 2016 | May 16, 2017 | Drip technique | 32.5 | 37.5 |
| **F2** | 1.5 | | Nov 27, 2017 | June 08, 2018 | | | |
| **F3** | 12 | 2018-2019 | Nov 04, 2018 | June 06, 2019 | | | |

**3  Database**
**3.1 Field datasets**
The field datasets consist of automatic measurements of soil moisture and weather data in addition to punctual surveys for
surface roughness, biomass, vegetation water content, canopy height, green leaf area index and cover fraction. Table A1 in
the appendix summarizes the details of the 26, 18 and 16 field campaigns carried out during 2016-2017, 2017-2018 and
2018-2019 seasons, respectively.

### 3.1.1 Soil moisture

SSM is automatically measured every 30 min using Time Domain Reflectometry sensors (TDR), (Campbell Scientific
CS616) using two sensors buried at a depth of 5 cm: one under the drippers and another one between. The average is
computed in order to get a representative SSM value of the field. In addition, similar sensors are buried for RZSM measuring
at 25 and 35 cm of depth over F1 and F3 while one sensor is buried at 30 cm over F2 by lack of additional sensor. Figure 2a
illustrates an example of TDR sensors at different depths.

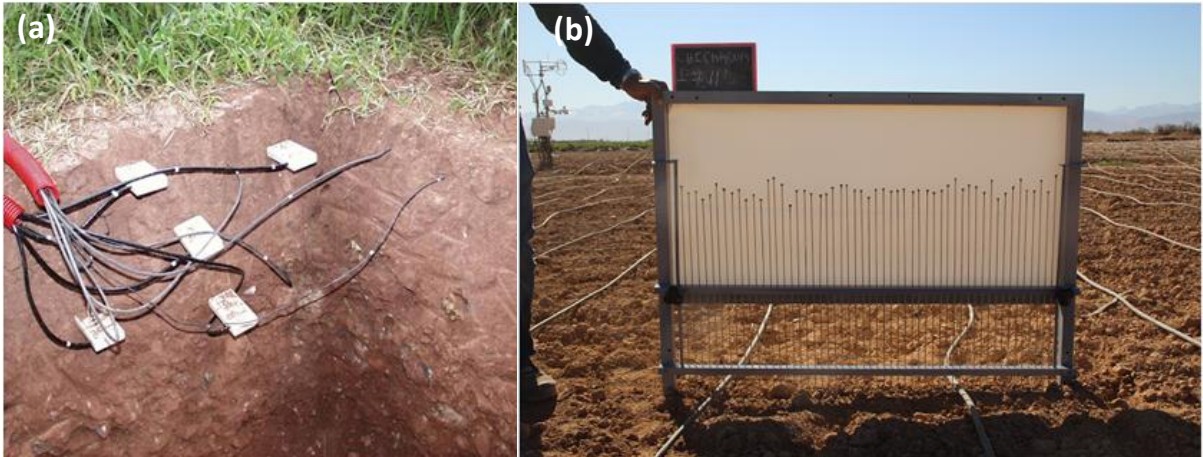

**Figure 2. Examples of (a) TDR sensors installed at different depths and b) a pin profiler picture taken over one of the ploughed field with drip irrigation tubes installed.**

TDR sensors are calibrated using the gravimetric technique. The calibration is done during 2016-2017 season using samples
taken from the first 5 cm from both fields F1 and F2 and then the calibrated equation is applied to F1, F2 and F3 data as the
soil characteristics are similar and the same sensors are used. For that purpose, an aluminum core of 392.5 cm$^3$ is used to
collect samples at the TDR installation depths. Three samples are collected per day and per field during five days chosen
with different soil moisture conditions in order to cover a wide range of values (0.08 to 0.33 m$^3$/m$^3$). A linear regression is
established between the volumetric water content and the square root of the TDR time response (named $\tau$ in second) as
follow:

$$SSM = a_{TDR} * \sqrt{\tau} + b_{TDR} \tag{1}$$

The calibrated values using data of both fields are $a_{TDR} = 0.275\ \text{m}^3/\text{m}^3/\text{s}^{0.5}$ and $b_{TDR} = -1.154\ \text{m}^3/\text{m}^3$. Figure 3
illustrates the calibration results with all the samples displayed. The statistical metrics are: correlation coefficient R = 0.97,
Root Mean Square Error RMSE = 0.018 m$^3$/m$^3$ and no Bias. When considering both fields separately, the results for (F1, F2)
are R = (0.90, 0.94), RMSE = (0.023, 0.01) m$^3$/m$^3$ and Bias = (-0.002, 0.003) m$^3$/m$^3$.
The calibrated equation is also applied for the RZSMs assuming that the soil properties are the same at different depths.
Figure A2 in appendix A illustrates an example of an RZSM time series over F1.

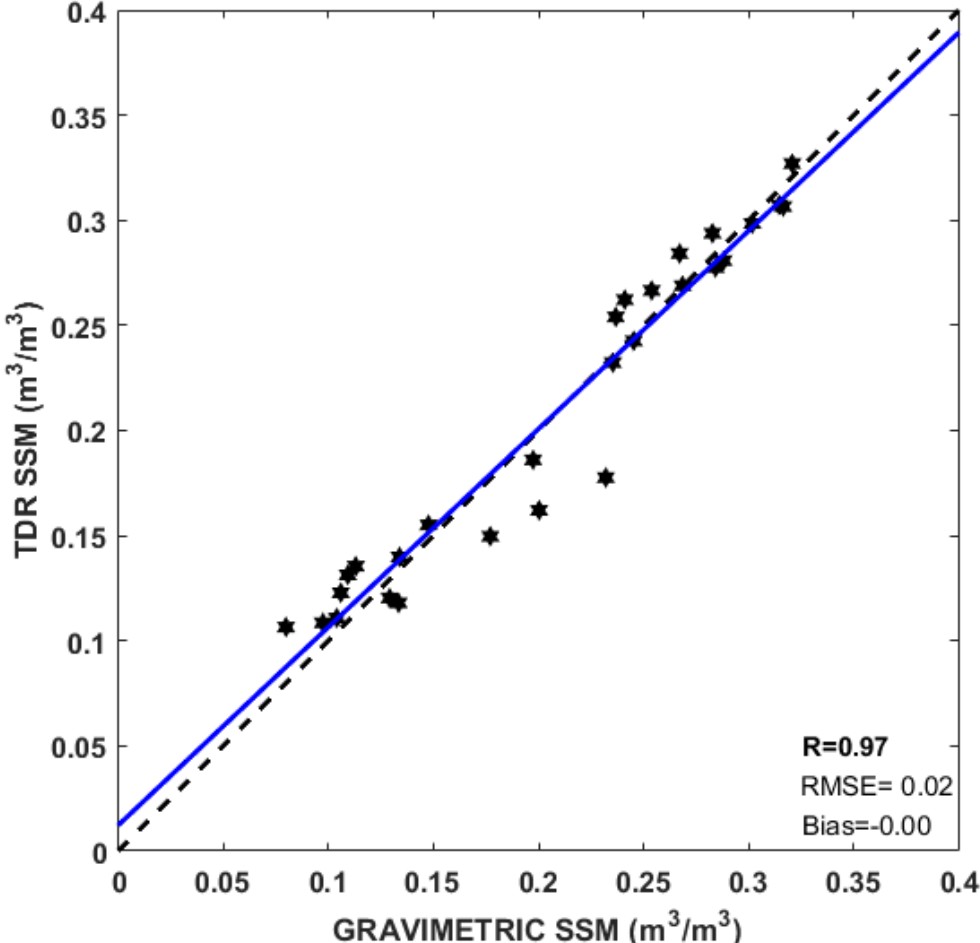


**Figure 3. Surface soil moisture measured by TDR versus gravimetric measurements using samples collected over both fields F1**
**and F2 during 2016-2017 growing season. The blue solid line is the linear regression and the dashed line is Y=X.**
**3.1.2 Surface roughness**
Surface roughness characterizes the micro variation of the ground surface elevation within a given area/field (Allmaras et al.,
1966). It affects particularly the SAR signal and to a lesser extent the visible and near infrared (Girard and Girard, 1989).
The two parameters that characterize the surface roughness are the root mean square height ($h_{rms}$) and the correlation length
(L). $h_{rms}$ provides a vertical descriptor of ground roughness by measuring the elevation of the surface along one or more
observation lines and calculating the standard deviation of the recorded values. The second parameter (L) corresponds to the
distance between measurements from which the heights between points are statistically independent. This parameter
provides a horizontal description of the ground surface roughness, more specifically the organizational structure and spatial
continuity of the microtopography (Nolin et al., 2005). Over the 3 studied fields, measurements of the surface roughness are
taken during the first stage of wheat (from emergence to early tillering) when the ground is not totally covered by the
canopy. We used a pin profiler of 1 m length, composed of a set of 53 metal needles of equal length every 2 cm (Fig. 2b). 16
sample pictures are taken per field and per date including eight pictures parallel and eight pictures perpendicular to the rows
direction. The pictures are taken using Canon 6EOS 600D equipped with TAMRON lens (Model A14).
The images are processed in MATLAB based on the detection of the top position of each needle. $h_{rms}$ and L are computed
from the auto-correlation function and then the average per direction, per field and per date is computed. For illustration, Fig.
4 shows the time series of $h_{rms}$ and L parameters computed separately for each direction for F1 and F2 during the season
2017-2018 while the average value per season are summarized in Table 2 for F1, F2 and F3.
**Table 2. Average values of the roughness parameters (8 samples are gathered per field and per direction).**

| | | F1 | | F2 | | F3 | |
|---|---|---|---|---|---|---|---|
| | | $h_{rms}$ (cm) | L (cm) | $h_{rms}$ (cm) | L (cm) | $h_{rms}$ (cm) | L (cm) |
| **2016-2017** | Parallel | 0.92 | 5.02 | 1.19 | 5.77 | | |
| | Perpendicular | 1.34 | 5.88 | 1.19 | 5.8 | | |
| | Average | 1.13 | 5.45 | 1.19 | 5.78 | | |
| **2017-2018** | Parallel | 0.89 | 5.44 | 1.1 | 5.88 | | |
| | Perpendicular | 1.16 | 7.4 | 1.12 | 6.6 | | |
| | Average | 1.02 | 6.42 | 1.11 | 6.24 | | |
| **2018-2019** | Parallel | | | | | 0.83 | 6.54 |
| | Perpendicular | | | | | 0.96 | 7.32 |
| | Average | | | | | 0.89 | 6.93 |


Based on the range of $h_{rms}$ measurements (0.83< $h_{rms}$ <1.35), it can be clearly seen that the fields are characterized by a
slightly rough or smooth surface, which is the general case of disk tilling fields. After sowing, a slight change is observed at
the start of the crop season (December 28, 2017, see Fig. 4). At that time, the soil has just been prepared for sowing and rows
are directly exposed to rain. The fact that the rows are still visible in the field also explains the differences observed between
both directions early in the season. This anisotropy disappeared quickly with irrigation, rainfall and plant growth. $h_{rms}$ and L
are almost constant from early January onwards. Indeed, it has been shown that after sowing, roughness is affected by very
limited temporal variations (Bousbih et al., 2017) as no soil works occur after sowing. It is usually kept constant during the
crop season (El Hajj et al., 2016; Gherboudj et al., 2011; Gorrab et al., 2015; Ouaadi et al., 2020).

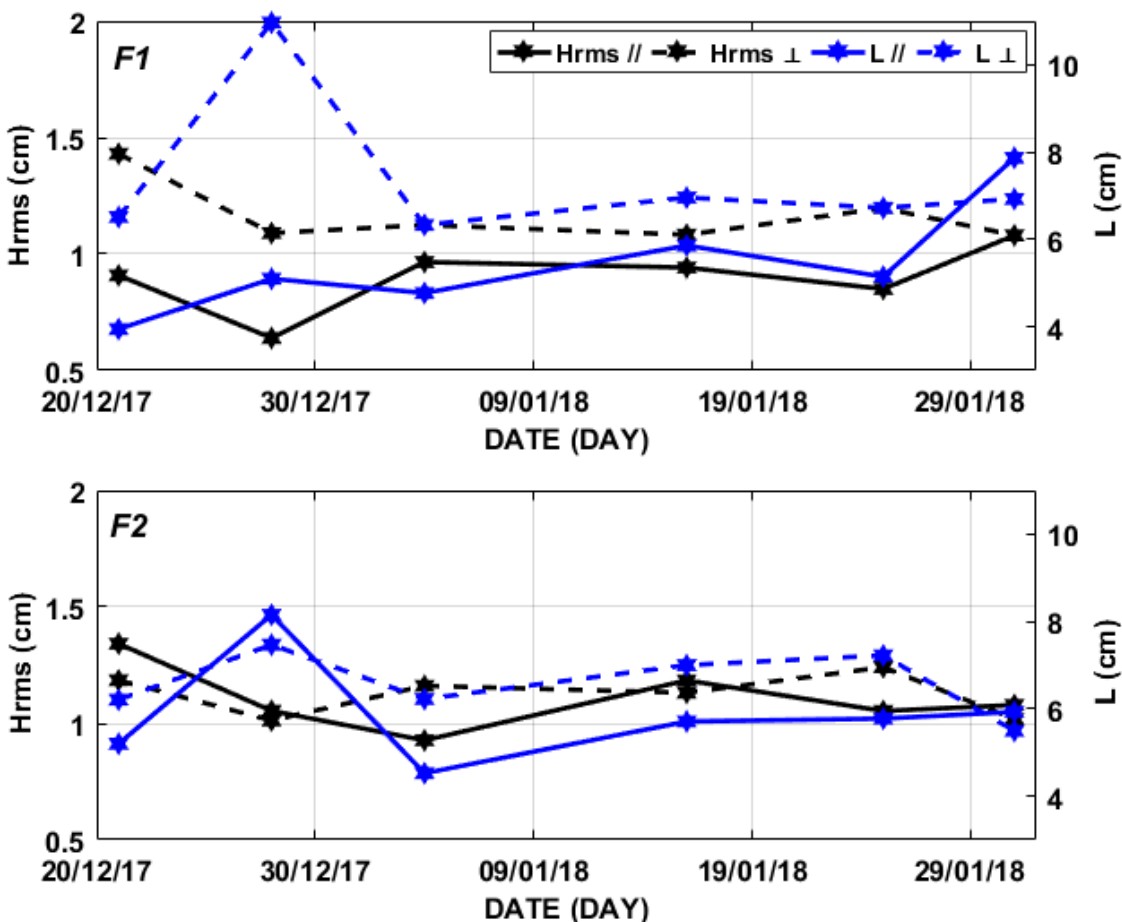


Figure 4. Time series of $h_{rms}$ and L computed from parallel and perpendicular measurements separately for F1 and F2 during the season 2017-2018.

### 3.1.3 Biomass and water content

Biomass and water content are two biophysical parameters of crucial importance in different agricultural applications including particularly plant stress monitoring, radar backscattering response, crop yield and evapotranspiration modeling. Within each field, eight samples are collected once a week/two weeks during the growing season. The samples are chosen randomly so that the average is representative of the plot. A quadrates of an area of 0.0625 $m^2$ is used for the sampling (Fig. 5). The samples are weighed first in the field to get fresh above-ground biomass (FAGB). The corresponding above ground biomass (AGB) expressed in kg of dry matter by $m^2$ is determined at the laboratory by drying the samples in an electric oven at 105°C for 48 hours. The vegetation water content (VWC) is thus computed as the difference between FAGB and AGB (Gherboudj et al., 2011).

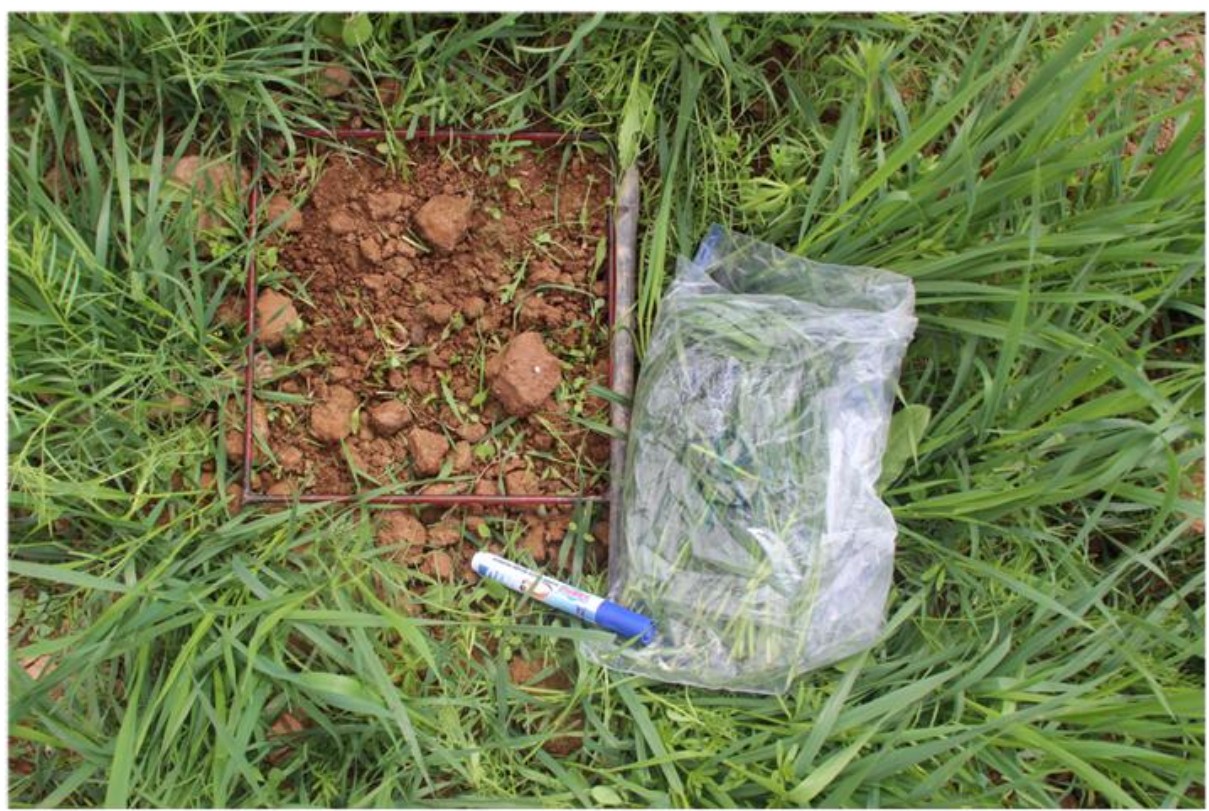


**Figure 5. Photo taken during a measurement campaign illustrating a sample of above-ground biomass measurement.**

**3.1.4 Canopy height, green leaf area index and cover fraction**
Canopy height (H), green leaf area index (GLAI) and cover fraction ($F_C$) are measured every week during the growing
season. Values from eleven different places are averaged and considered as a representative measure of the field. H is simply
measured using a measuring tape while GLAI and Fc are computed by processing hemispherical photos (Fig. 6b) using
MATLAB software following the method described in Duchemin et al. (2006) and Khabba et al. (2009). The eight photos
per date and per field are taken using a camera Canon 6EOS 600D with SIGMA 4.5 mm F2.8 EXDC circular fisheye HSM
(Fig. 6a). Photos are taken in optimal lighting conditions to avoid shadow effects and over-exposure phenomena which make
classification more difficult. The algorithm is based on the binarization of the hemispherical images by thresholding a
greenness index. Next, the useful part of the images is extracted by masking the operator and the high viewing angles ($> 75°$)
(Fig. 6c). Finally, the ground-covered area is extracted on concentric rings associated with fixed viewing angles and the
average of all pictures is the field GLAI. Using the same process, Fc is calculated as the ratio of the vegetation pixels number
to the pixels total number.

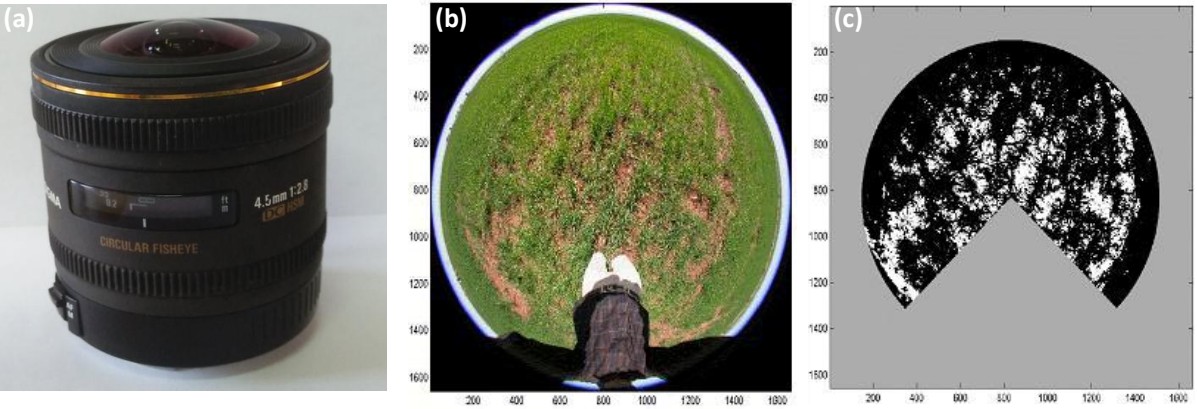


**Figure 6. (a) the 4.5 mm F2.8 EXDC circular fisheye HSM, (b) hemispherical photo and (c) result of the processing after binarization and after masking the operator and the high viewing angles (> 75°).**

### 3.1.5 Irrigation and weather data

F1, F2 and F3 are irrigated using the drip technique. Irrigation quantities are determined by the farmer by estimating the daily evapotranspiration under standard conditions (ETc) in the region computed using the FAO-56 model simple approach (Allen et al., 1998). The cumulative $ET_c$ for a given period (usually one week) is applied during one or more events per week depending on the farmer's constraints (e.g. availability of workforce) and on the weather conditions (e.g. occurrence of rain). The irrigation pipes are spaced by 0.7 m while the distance between the drippers along the pipe is 0.4 m. Over F1 and F2, the flowrate of each dripper is 7.14 mm/hour. The irrigation takes place about 105 min (12.53 mm). A flowmeter mounted downstream of a valve allowed an accurate collection of irrigation volumes. F2 and F3 are irrigated according to FAO recommendations while F1 is stressed voluntarily. The stress involved in F1 is during the first season (2016-2017) only. By contrast, the 2017-2018 season was wet so that there is no clear stress observed on the field. The irrigation dates and amounts over F1 and F2 during both seasons are made available throughout this database while irrigation over F3 are not available.

The weather data including precipitation, air temperature, relative humidity, solar radiation, wind speed and direction are collected by an automatic weather station installed over an alfalfa field near the studied fields (Fig. 7). The weather station provides continuously meteorological data every 30 min. The sensor Campbell CS215 is used to measure the air temperature and the relative humidity (Fig. 7). The global solar radiation and the wind direction and speed are measured using Campbell SKP215 and Campbell windsonic4, respectively. The precipitation are measured using the Rain Gauge (Campbell SBS500) shown in Fig. 7.

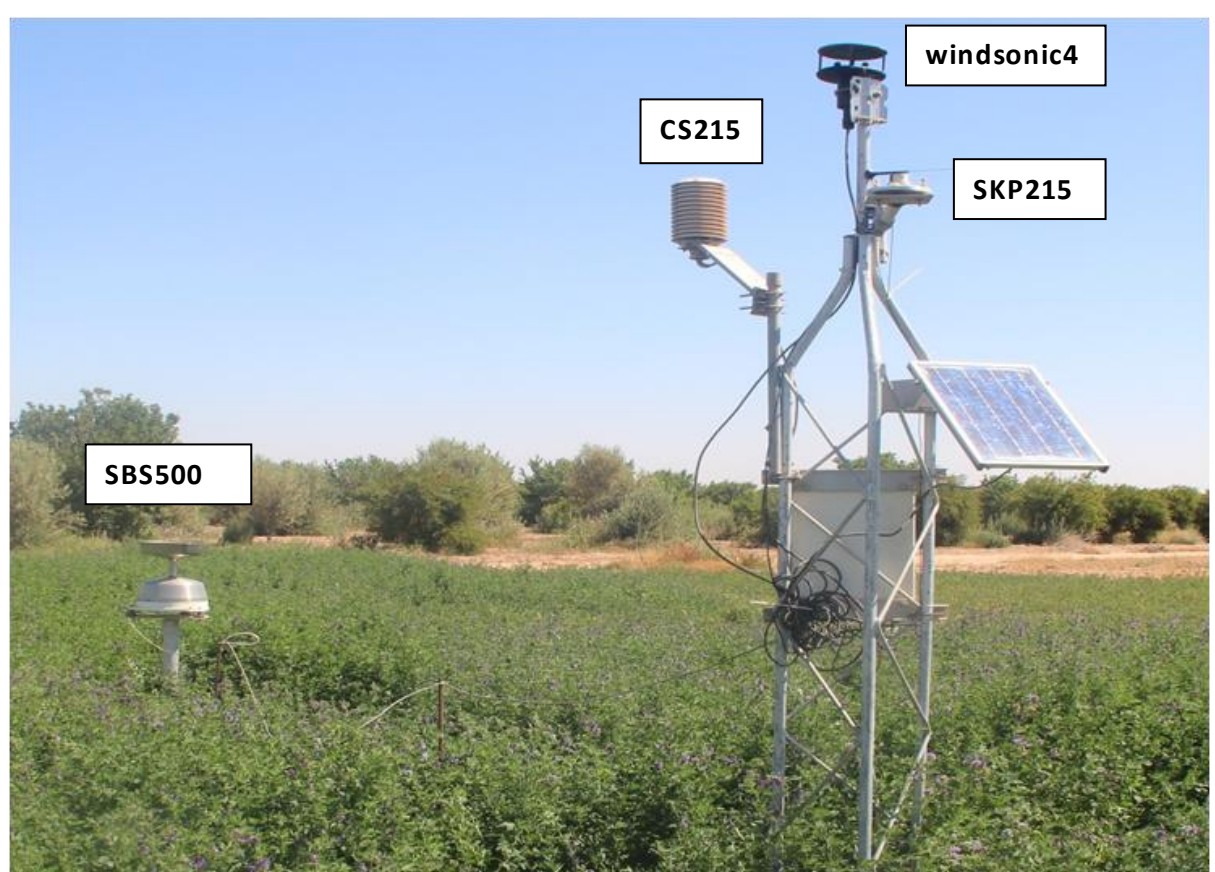


**Figure 7. Automatic weather station installed over an alfalfa field near F1, F2 and F3.**

**3.2 Remote sensing datasets**
**3.2.1 Sentinel-1**
Sentinel S1A and S1B are earth observation satellites developed for the Copernicus initiative and launched by the European
Space Agency on April 2014 and 2016, respectively. During full operation, S1A and S1B are maintained in the Near-polar
Sun-synchronous orbit at 693 km altitude, phased 180°, providing a revisit time of six days (Torres et al., 2012). S1 is a
synthetic aperture radar operating at C-band with a frequency of 5.33 GHz, mapping the entire world in 175 orbits per cycle.
The main operational imaging mode is the Interferometric Wide-swath mode (IW). IW acquires data with a wide swath of
250 km with high geometric (azimuth resolution 20 m and ground range resolution 5 m) and radiometric resolution (Mission
and Services, 2012). IW mode supports operation in single and dual polarization (HH, VV, HH/HV and VV/VH) and covers
a range of incidence angles between 31° and 46°. The product is composed of three Sub-Swath acquired in TOPSAR
imaging technique which significantly reduces the scalloping effect (Zan and Guarnieri, 2006).
Level 1 products are systematically processed and available within 24 hours, free of charge from the Sentinel-1 Data Hub
website (https://scihub.copernicus.eu). The website provides data under two types of products: GRDH (Ground Range
Detected High resolution) and SLC (Single Look Complex).
In this database, 561 GRDH and SLC products are processed (Table 3). Among them, 124 images are acquired over F3
during 2018-2019 growing season and 437 over F1 and F2 from October 01, 2016, to July 31, 2018, along the ascending
#118 (221 images) and descending #52 (216 images) relative orbits. This period includes two agricultural seasons in addition
to the summer period.
**Table 3. Characteristics of the sentinel-1 products processed over the three fields for the monitored periods**

| Field | Season | Relative Orbit Number | Incidence angle | Relative Orbit | Overpass time | Product | Number of images |
|-------|--------|----------------------|-----------------|----------------|---------------|---------|------------------|
| **F1 and F2** | October 2016 - July 2018 | 118 | 45,6° | Ascending | 18:30 | GRDH | 112 |
| | | | | | | SLC | 109 |
| | | 52 | 35,2° | Descending | 06:30 | GRDH | 110 |
| | | | | | | SLC | 106 |
| **F3** | November 2018 - May 2019 | 118 | 45,6° | Ascending | 18:30 | GRDH | 32 |
| | | | | | | SLC | 31 |
| | | 52 | 35,2° | Descending | 06:30 | GRDH | 31 |
| | | | | | | SLC | 30 |

*Backscattering coefficient*
GRDH products are provided by ESA with a square pixel size and contains only the intensity information. The
backscattering coefficients are extracted using the Orfeotoolbox (CNES, 2018). The processing procedure consists of three
steps (Frison and Lardeux, 2018) :
1. *Thermal noise removal*: SAR product contains not only the useful signal but also the unwanted noise disturbing the
information contained in the intensity images, especially when the backscattered power is low. The thermal noise is
an additive noise. The compensation of this noise can be performed by subtracting the scaled noise power using the
calibrated noise vectors provided by ESA.
2. *Calibration*: The "calibration" step aims to convert the digital accounts into a physically interpreted parameter: the
backscattering coefficient. A calibration vector included in the GRDH products contains the necessary information
to convert the digital values to the backscattering coefficient.
3. *Terrain correction*: S1-SAR data are sensed with viewing angle greater than zero which induces distortion in the
products because of the lateral viewing geometry. The "Terrain corrections" module is used to compensate these
distortions and get as much possible images with the real geometric representation. The images are projected on the

266        Earth's surface using a Digital Elevation Model (DTM). The DTM SRTM (Shuttle Radar Topography Mission) of

267        30 m of resolution is used according to the method described in Small and Schubert (2008).

SAR images are affected by the speckle noise, which is mainly due to the relative phase of individual scatters within a
resolution cell. Many filters have been developed to remove the speckle noise although the best filter is the spatial average.
The presented database is generated using a simple average per field of 120, 121 and 1100 pixels for F1, F2 and F3,
respectively with a mean standard deviation of around 1.55 dB. In order to visualise data dynamics, backscattering
coefficients are converted into dB.
*Interferometric coherence*
Sentinel-1 SLC products are provided in slant-range geometry. It contains three sub-swath images IW1, IW2 and IW3. Each
sub-swath is composed by nine bursts with black-fill demarcation. By contrast with GRD, both intensity and phase
information are kept. The phase information is used for the computation of interferometric coherence. SAR interferometry
consists of correlating two images acquired from two positions in space slightly separated from each other (with two radars
mounted on the same platform) or at different times by exploiting repeated orbits of the same satellite such as for Sentinel-1.
Thanks to its high temporal resolution (six days per orbit), the interferometric coherence is computed from two consecutive
acquisitions of the same orbit.
The interferometric coherence, given by the Eq. (2), for a local neighborhood of N pixels, is generated by cross multiplying,
pixel by pixel, the first SAR image $z_i$ with the complex conjugate $z_i^{'*}$ of the second (Bamler and Hartl, 1998; Touzi et al.,

283 1999).


$$\rho = \frac{\sum_{i=1}^{N} z_i \cdot z_i^{'*}}{\sqrt{\sum_{i=1}^{N} |z_i|^2 \cdot \sum_{i=1}^{N} |z_i'|^2}} \tag{2}$$

The interferometric coherence $|\rho|$ varies between zero (incoherence) and one (perfect coherence). The interferometric
coherence is related to the movements of the scatterers within a given canopy. It decreases (loss of coherence) in the case of
dense vegetation while high values are obtained over bare soils. Loss of coherence could be caused by temporal interval
between acquisitions, orbit errors, vegetation development/movement or processing errors. The random dislocation of
scatters because of the weather (wind and rain) or the plants growth is the main cause of the temporal decorrelation.
Sentinel application platform SNAP is used to compute the interferometric coherence from S1-SLC products in five steps
(Veci, 2015):
1. *Apply-Orbit-file*: This module is applied for a better estimation of the position and speed of the satellite using the

294         orbit state vector. Preliminary, a predicted orbit state vector is contained in the metadata but it is not accurate. The

295         precise orbit is made available one month after data acquisition at the later. For this reason, the automatic download

296         in SNAP is used in order to update the orbit state vectors.

2. *Back-geocoding*: The two images need to be co-registered. One of the images is the master and the other is the slave. This step ensures that each pixel of the slave image is aligned with the corresponding pixel in the master image so that both pixels contain contributions from the same target. The DEM is required for "*Back-geocoding"* step, SNAP allows either to enter it manually or to download it automatically.

3. *Coherence*: This module in SNAP allows the computation of the interferometric coherence between the two images for a given local neighborhood. In order to get a square pixel of 13.95 m, azimuth* range are fixed to 3*15 in the processing.

4. *TOPSAR-Deburst*: The black-fills in between bursts are deleted separately for both polarization images (VV and VH).

5. *Terrain-Correction*: Finally, the processed images are projected on the earth surface using a DEM.

### 3.2.2 Sentinel-2 NDVI

Sentinel-2 optical satellites S2A and S2B are launched by ESA in June 2015 and March 2017, respectively. They are placed in opposition on the same orbit at an altitude of 800 km. Sentinel-2 provides data every 5 days with a width of 290 km and a resolution of 10 to 60 m according to spectral bands (13 bands) ranging from visible to the medium infrared. The National Centre for Space Studies (CNES) provides Level-2A products atmospherically corrected free of charge via the PEPS platform (https://peps.cnes.fr/) or the Theia website (https://theia.cnes.fr/). Data are corrected from atmospheric effects by the Center for the Study of the Biosphere from Space (CESBIO) using the MAJA chain (Hagolle et al., 2015). The atmospheric corrections are performed in three steps:

1. The satellite top-of-atmosphere (TOA) reflectances are corrected from the absorption by the atmospheric gas molecules using the absorption part of the Simplified Model for Atmospheric Correction (SMAC) method by Rahman et al., 1994. The concentrations of the ozone, the oxygen and the water vapor are obtained from satellite data (ozone) and meteorological data (water vapor, pressure).

2. The detection of the clouds (and cloud's shadows) is based on the multi-temporal cloud detection method proposed by Hagolle et al., 2010 .

3. The estimation of the aerosol optical thickness (AOT) relies on a hybrid method merging the criteria of a multi-spectral method with the multi-temporal technique developed initially for the VENµS satellite mission by Hagolle et al., 2010. The AOT is used along with the surface altitude, the viewing geometry and the wavelength in the parameterization of look-up tables for the conversion of TOA reflectances already corrected in step "1" into surface reflectances. The look-up tables are provided by the successive orders of scattering code (Lenobel et al., 2007) used in the modeling of molecular and aerosol scattering effects. A different look-up table is computed for each aerosol model.

Data are downloaded from the Theia site. Among the available products, only the products non-covered with clouds are used corresponding to ten, twenty-five and twenty-six images for 2016-2017, 2017-2018 and 2018-2019 agricultural seasons, respectively. Please note that during the season 2016-2017, only S2A was in the orbit which explains the limited number of images (10). Next, the Normalized Difference Vegetation Index (NDVI) corresponding to each pixel is computed from band 4 and 8. An average per field is used to compute the time series of each field.

## 4 Data analysis

### 4.1 Vegetation variables

In this section, the relationships between the different variables (GLAI, FAGB, AGB, VWC and H) that characterize the vegetation growth and development are firstly investigated. These relationships are extensively used for different applications such as the calibration of backscattering models and the development of retrieval approaches (Chauhan et al., 2018). Several land surface or crop model relies on empirical relationships to predict Fc or H as well (Bigeard et al., 2017; Castelli et al., 2018). Other agricultural models compute AGB from GLAI using linear or polynomial relationships (Major et al., 1986; Petcu et al., 2003). Figure 8 displays the resulting relationships using data from F1 by selecting only the 2016-2017 season for illustration purposes. These relationships are computed separately based on the data recorded before and after the peaks of GLAI and FAGB.

The nature of the relationship changes depending on the structure (biomass variables) or on the greenness of the plant (GLAI). The biomass variables (FAGB, AGB and VWC) and H increase up to the biomass peak. Afterwards, a reverse evolution can be observed characterized in particular, by a decorrelation between FAGB/VWC and AGB. This is mainly related to the senescence process of the vegetation; the leaves begin to dry progressively with the start of the grain filling, so that the Sapflow (water, carbohydrates, proteins and mineral salts) migrates to the heads at the top of the plant (Farineau and Morot-Gaudry, 2018). Indeed, VWC and AGB are highly correlated until vegetation peak (the correlation coefficient R = 0.94 before the peak and R = -0.20 afterwards) while FAGB being dominated by the plant water content is highly correlated with VWC during the whole crop season (R=0.99 before the peak and R = 0.98 afterwards). Likewise, H is highly correlated to FAGB, VWC and AGB until vegetation peak (R > 0.97) when H remains at its maximum value while AGB continue to increase with grain filling and VWC and FAGB decrease because of the vegetation drying. The relationship of these variables (FAGB, AGB, VWC and H) with GLAI and Fc is quite different. The curves are of a parabolic shape with a maximum reached around the GLAI peak. A timing shift between the peaks of GLAI and FAGB is observed. This is probably related to the senescence of the lower leaves, which leads to an earlier drop of GLAI than of FAGB. Between the peaks of GLAI and FAGB, GLAI decreases while i) AGB and H increase and ii) FAGB increases slightly while VWC is almost constant. After the FAGB peak, AGB goes on increasing due to grain filling while the VWC decreases due to drying of the plant. FAGB which is the sum of AGB and VWC is almost constant.

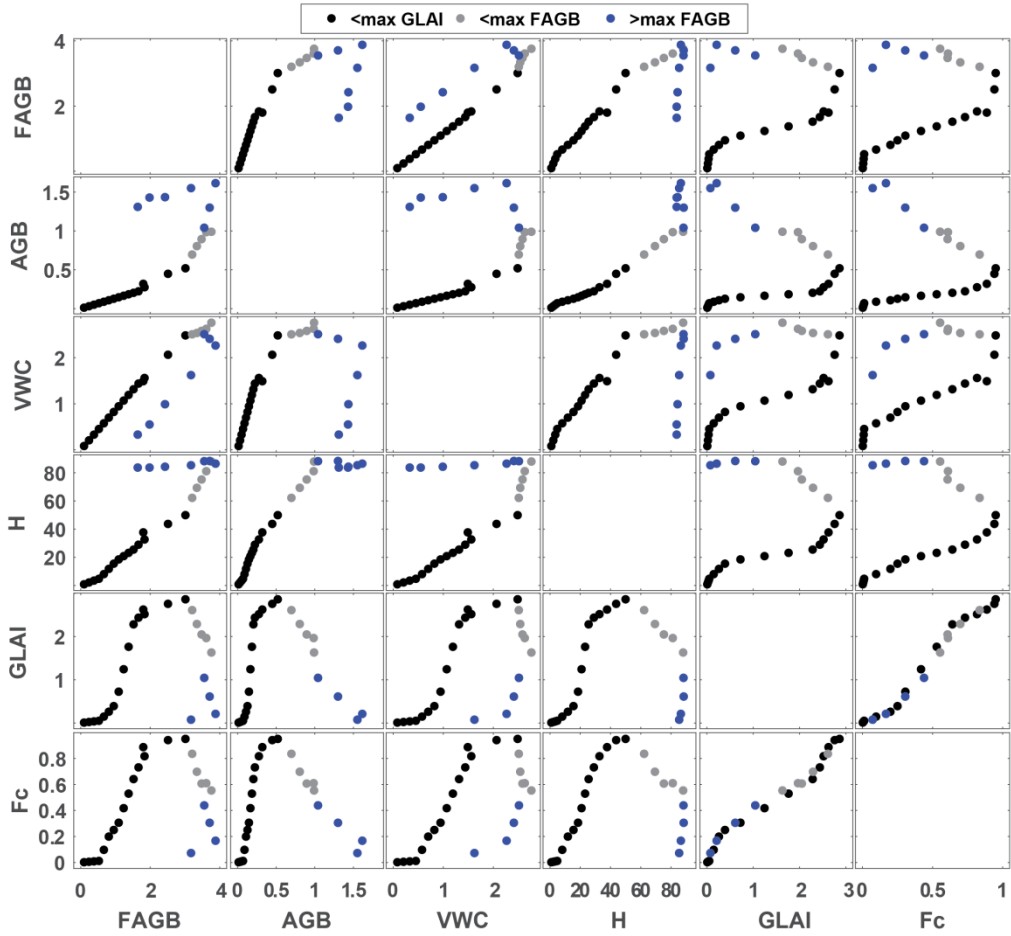

**Figure 8. Scatterplots of the relationships between wheat measured variables: FAGB, AGB, VWC, H, GLAI and Fc. Data are presented separately using the maximum of GLAI and FAGB as thresholds: data <max GLAI are in black, data < max FAGB (and > max GLAI) are in grey and data > max FAGB (and > max GLAI) are in blue.**

## 4.2 Radar data

The time series of the backscattering coefficient, the polarization ratio and the interferometric coherence are analyzed here for two agricultural seasons and a summer period on F1 and F2 and at two incidence angles (35.2° and 45.6°).

### 4.2.1 The backscattering coefficient

Figure 9 displays the time series at 45.6° over F2 for illustration purposes: a) backscattering coefficient at VV polarization ($\sigma^0_{VV}$); b) backscattering coefficient at VH polarization ($\sigma^0_{VH}$) as well as wheat phenological stages; c) SSM, air temperature,

irrigation and rainfall. Figures A3- A5 in appendix A show the same time series over F1 and F2 at 35.2° and F1 at 45.6°, respectively. The backscattering coefficients reveal a strong seasonal signal with two cycles. The first cycle takes place from sowing to the heading stage and the second from heading to harvest with the minimum reached around the heading stage. The highest values at 35.2° are observed in the first cycle, while at 45.6°, σ° is higher on the second peak. The maximum values of $\sigma^0_{VV}$ reached the same value for F1 and F2 while higher values are observed on F2 at VH. $\sigma^0_{VV}$ is more sensitive to soil moisture variation until mid-January, corresponding to the tillering stage, when the soil is not yet fully covered by vegetation. Although it is agreed that the signal during this period is governed by the dynamics of soil moisture, its behavior differs from one site to another giving the difference in soil hydric conditions and surface roughness. After this period, the signal behavior is similar to the profiles obtained by Cookmartin et al. (2000), El Hajj et al. (2019), Nasrallah et al. (2019) and Veloso et al. (2017). It decreases gradually from the early tillering until the heading stage (around March 13) by about 10 dB on F2 and 5 dB on F1 because of the attenuation by the canopy during the development of the stems (extension stage) (Cookmartin et al., 2000; Mattia et al., 2003; Picard et al., 2003; Wang et al., 2018). Obviously, the attenuation is more important at VV polarization because of the vertical structure of wheat (stems) in line with the results of (Fontanelli et al., 2013; Picard et al., 2003; Wang et al., 2018). The response of $\sigma^0_{VH}$ to SSM variation and canopy attenuation is lower than for $\sigma^0_{VV}$. After the heading stage, the signal starts to increase again. This is clearer on F2 than F1 and at 45.6° than at 35.2°. The heading stage is the phenological stage of wheat when the spike or head starts emerging out from the leaf sheath. This change of the structure of the canopy shield the stems for the radar signal through the appearance of a thick, wet top layer composed of the heads. The C-band wavelength penetrates this layer only, resulting in increased volume scattering, while attenuation becomes low. This effect is stronger for F2 than for F1, at VH than at VV and at 45.6° than at 35.2°. This increase was first reported by Ulaby and Batlivala (1976). Subsequently, Ulaby et al. (1986) suggested that an additional term must be added to the traditional three-term model (vegetation volume diffusion, soil attenuation, and soil-vegetation interaction) to properly represent wheat backscattering after heading. Later on, similar behaviour has been observed and attributed to the appearance of the heads followed by the grain by numerous authors (Brown et al., 2003; El Hajj et al., 2019; Mattia et al., 2003; Patel et al., 2006; Veloso et al., 2017). The exceptional growing conditions on F2 during S2 is at the origin of the observed plateau of the backscattering coefficient which remains quite stable until harvest. This is due to a significant contribution of volume scattering which is a behavior that characterize a crop developing a random canopy structure in relation to the numerous and dense adventices as already highlighted (cf. picture Fig. A1 at appendix A).

The low variation observed on F1 during the 2016-2017 season is mainly related to the limited development of vegetation because of the triggered water stress. Likewise, the difference between the two seasons over F2 is related to a higher density of grown seeds and wetter conditions in the 2017-2018 season compared to 2016-2017 (the amount of rainfall during the growing season-from sowing to harvest-reached 167.23 mm in 2017-20118 while only 69.94 mm is recorded in 2016-2017). With the drying of the head layer, the backscattering decreases again at the end of the season to reach the lower observed values. Indeed, as the head layer dries, the vegetation becomes transparent to the signal. The soil is also dry at the end of the

season because irrigation is stopped. These low values remained until the first deep ploughing on July 11, when a sharp
increase is observed because of a drastic change of soil roughness. Hereafter, the signal is again stable until the seedling
preparation work for the next 2017-2018 season (November 22).

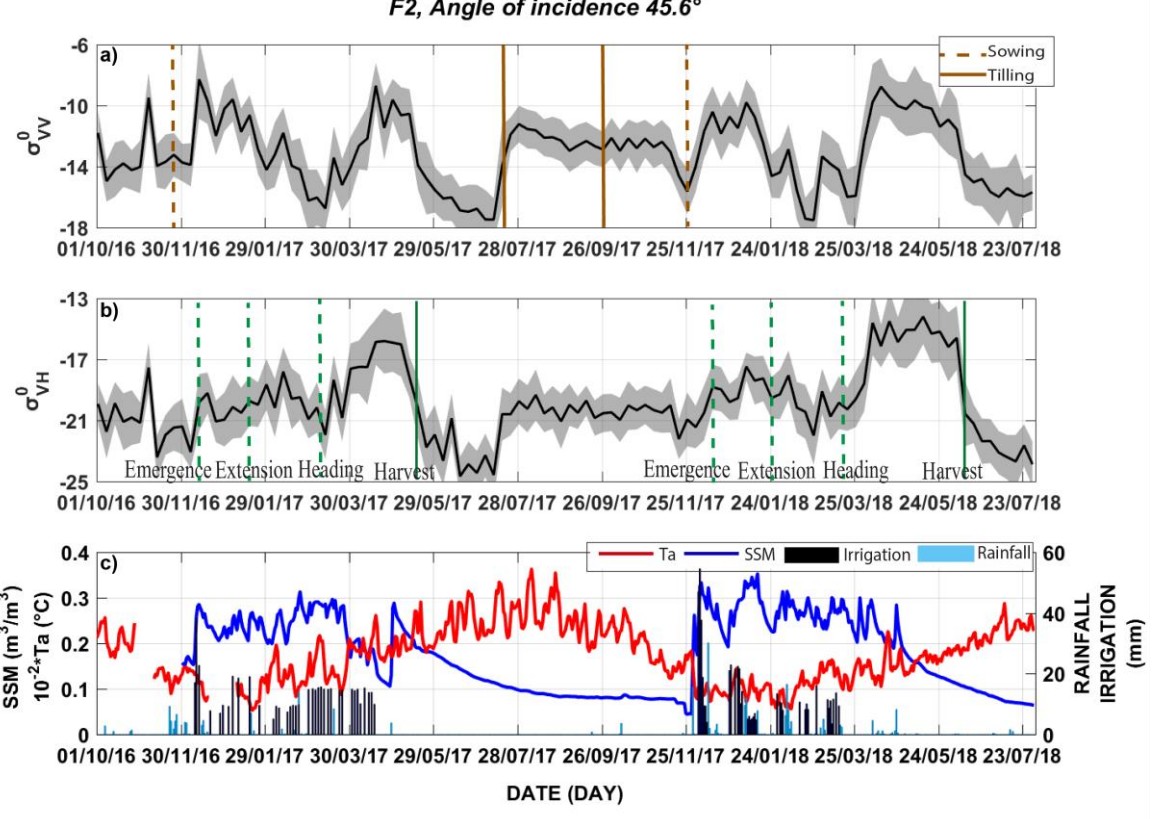

**Figure 9. Time series of the backscattering coefficient at VV (a) and VH (b) polarizations on F2 at 45.6° of incidence angle during**
**the period from October 01, 2016 to July 31, 2018. The tilling works and phenological stages of wheat are superimposed on**
**subplots (a) and (b), respectively. The air temperature, surface soil moisture (SSM), irrigation and rainfall are displayed on**
**subplot (c).**
**4.2.2 The interferometric coherence and the polarization ratio**
Figure 10 displays the time series at 45.6° over F2 of: a) interferometric coherence at VV ($\rho_{VV}$) and VH ($\rho_{VH}$) polarizations
together with sowing and tilling dates; b) polarization ratio (PR $= \sigma_{VH}^0/\sigma_{VV}^0$), as well as wheat phenological stages; c)
Sentinel-2 NDVI and measured GLAI; d) Measured FAGB, AGB, VWC and H. Likewise, Fig. A6- Fig. A8 in appendix A
display the time series over F1 and F2 at 35.2° and F1 at 45.6°, respectively. The time series of $\rho_{VV}$ and $\rho_{VH}$ follows a
similar evolution. Before sowing, coherence is at its highest value corresponding to 0.9 for $\rho_{VV}$ and 0.7 for $\rho_{VH}$ (Fig. 10a).
These values express a dominance of coherent scattering, corresponding to response of bare soils composed of big rocks.
Indeed, during the summer, the plots are subjected to deep ploughing which yields big clods that resist any change in surface

structure caused by climatic factors such as wind or rain. The second tilling breaks up the clods for the next seeding. Soil works and farming activities induce a large decrease in coherence in line with the observation of Wegmuller and Werner (1997). The surface roughness is a main parameter that influences not only the amplitude at C-band but also the phase. Indeed, abrupt drops are observed around each sowing events and tilling works (brown vertical lines on Fig. 10a).

After sowing, the evolution is similar to the profiles obtained by Blaes and Defourny (2003) and Engdahl et al. (2001). The interferometric coherence increases from 0.15 to 0.7 and then starts to decrease slightly from the emergence of wheat, becoming almost constant after stem extension with values < 0.3 corresponding to the noise level. Indeed, using the ERS–Envisat Tandem mission, Santoro et al., (2010) demonstrated that coherence measurements of vegetated fields are always below the level of bare soils coherence. Actually, the interferometric coherence is known to decrease exponentially with wheat growth (Lee et al., 2012). Vegetation growth and random dislocation of scatters cause a degradation of coherence (Blaes and Defourny, 2003; Engdahl et al., 2001; Wegmuller and Werner, 1997), especially under wind and rain effects. Between sowing and emergence, the observed variation is assumed to be related to the installation of irrigation drippers that took place up to two weeks after sowing. The changes that occur between the harvest and the first tilling could be attributed to livestock grazing, a common practice in the region after wheat harvest, which could change the surface roughness.

The polarization ratio (PR) is closely related to the biomass dynamic. Both are increasing from emergence to heading and then start to decrease until harvest. The maximum timing is around middle of April. The significant differences in biophysical parameters between F1 and F2 is due to irrigation, as already highlighted for the backscattering coefficient time series. Likewise, the difference between the two seasons over F2 is related to a higher sowing density and wetter conditions in the 2017-2018 season compared to 2016-2017. As shown above (Fig. 8), the time series of FAGB and VWC are in line with AGB and H up to the peak of FAGB and then decrease together while AGB continues to increase and H remains at its maximum value. FAGB and VWC are dropping at the same time but 50 days after when compared to GLAI and NDVI and about 15 days before the backscattering coefficient.

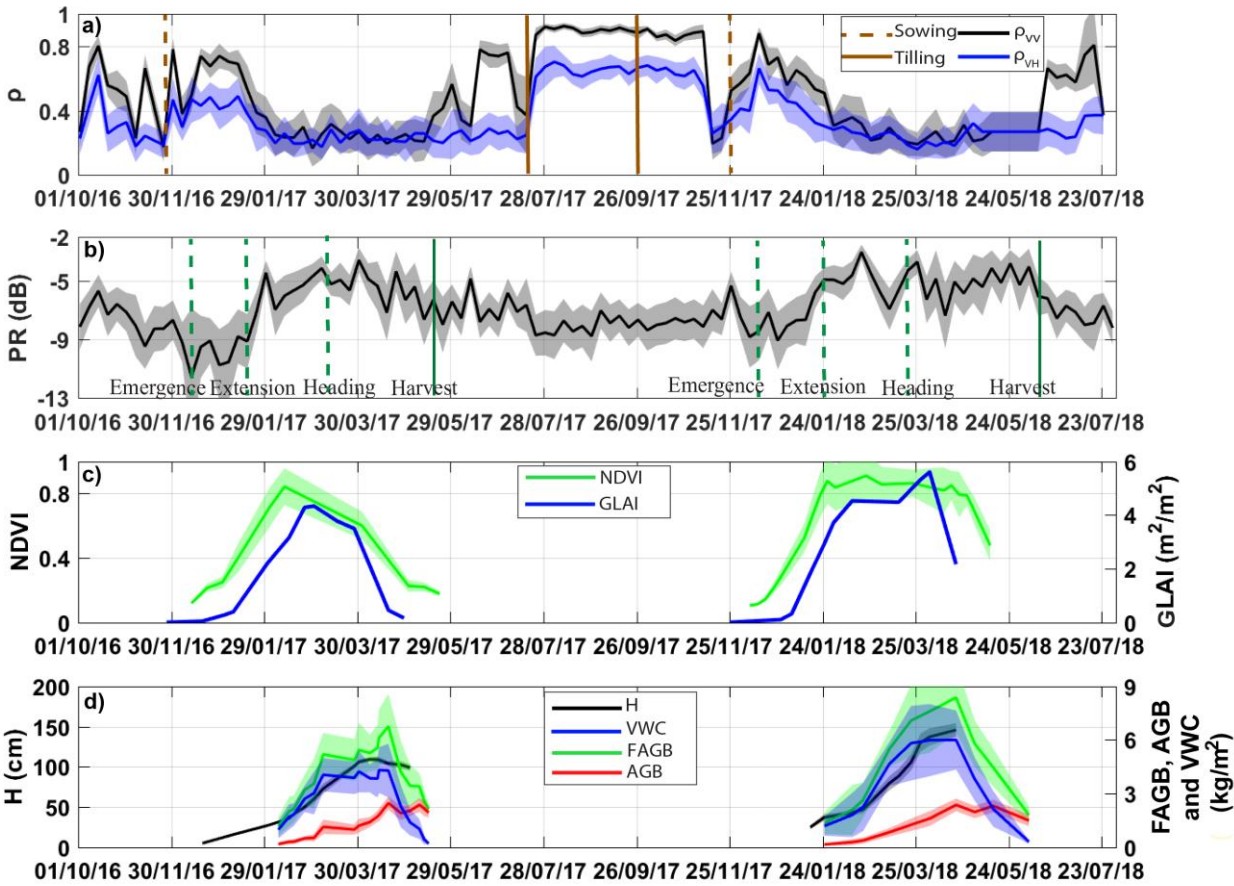

440

**Figure 10. Time series of the interferometric coherence at VV and VH polarizations (a) and the polarization ratio (b) on F2 at 45.6° of incidence angle during the period from October 01, 2016 to July 31, 2018. The tilling works and phenological stages of wheat are superimposed on subplots (a) and (b), respectively. NDVI and measured GLAI are displayed in subplot (c). Measured H, FAGB, VWC and AGB are plotted in subplot (d). Time series are presented by mean values (solid lines) and standard deviations (filled fields surrounding the solid lines).**

**4.3 Relationship between SAR data and vegetation variables**

The polarization ratio and the interferometric coherence have been shown to be related to vegetation growth. In this section, the relationships between PR, $\rho_{VV}$ and $\rho_{VH}$ and vegetation variables, including AGB, VWC, H, GLAI and NDVI are analyzed. Figure 11 displays the results at 35.2° of incidence angle and Fig. A9 in appendix A displays the results at 45.6°. H is used to illustrate the vegetation growth because its evolution is monotonic, so that data corresponding to before and after maximum development can be easily separated. The determination coefficient $R^2$ and the Spearman rank correlation Rs are superimposed on the subplots together with the fitting equations using the whole database. Overall, a good correlation has

been found between SAR variables (PR, $\rho_{VV}$ and $\rho_{VH}$) and AGB, VWC, GLAI and H. A hysteresis behavior is obviously observed for the vegetation variables with a non-monotonic dynamic (VWC, NDVI and GLAI). Using PR, the relationships are more scattered and characterized by lower saturation value. Although the range of variation of $\rho_{VH}$ is limited with regards to PR, the statistical metrics of the relationships between interferometric coherences and the vegetation variables are better than those obtained using PR. $\rho_{VV}$ exhibited better correlation to the vegetation variables than $\rho_{VH}$. With the exception of NDVI, Rs is always greater than 0.67. The best fit is obtained between $\rho_{VV}$ and H (Rs = 0.78 and $R^2$ = 0.65) with higher saturation value than the other relationships (~55% of H range which is about 77 cm). By contrast, a visual inspection of the Fig. 11 (d, i and n) shows that relationships with NDVI are poorer when using data of the whole growing season. The dispersion is strong along the season. Data before and after the maximum development can be distinguished, particularly, using $\rho_{VV}$ and to a lesser extent $\rho_{VH}$. Figure 11 (i and n) shows that a linear relationship exists between NDVI and SAR data using data before maximum development only, i.e. when the vegetation is still green. During the beginning of the season, the slope of $\rho_{VV}$-NDVI and $\rho_{VH}$-NDVI is low compared to the other vegetation variables. This is because the NDVI increases faster around the emergence of wheat while $\rho_{VV}$ is steel high because of the low vegetation cover fraction at this time. The hysteresis effect observed after the maximum of vegetation development is due to the senescence of the leaves when NDVI starts decreasing while $\rho_{VV}$ and $\rho_{VH}$ are stable at low values.

When considering SAR data at 45.6° of incidence angle (Fig. A9), a similar behavior to Fig. 11 is observed with AGB, VWC, H and NDVI. Same hysteresis and scattering are observed for NDVI although higher correlations are obtained. Similarly, $\rho_{VV}$ is better correlated to vegetation parameter than $\rho_{VH}$ and PR. By contrast, GLAI is better correlated with SAR variables than H. The PR-GLAI relationship is more scattered than at 35.2° while $\rho_{VV}$-GLAI has the best metrics (Rs = 0.82 and $R^2$ = 0.73) with a higher saturation value around 50% of the GLAI range (3 $m^2$ $m^{-2}$).

Unlike PR, the metrics at both 35.2° and 45.6° are stable for the relationships between $\rho_{VV}$ with AGB, VWC and H. By contrast, PR-GLAI is more stable than $\rho_{VV}$-GLAI at both incidence angles.

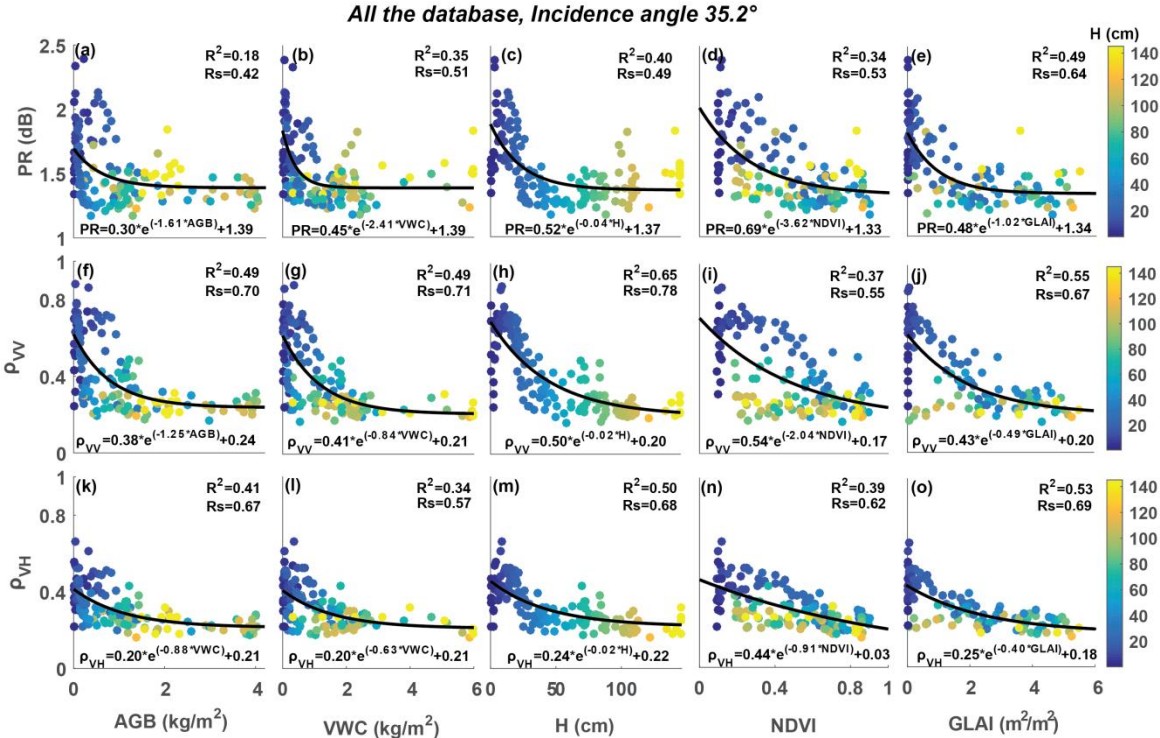

All the database, Incidence angle 35.2°

**Figure 11. Scatter plots of the relationships between PR, $\rho_{VV}$ and $\rho_{VH}$ and AGB, VWC, H, NDVI and GLAI at 35.2° angle of**
**incidence. The entire database is used from the three fields (F1, F2 and F3). H is used to monitor the evolution during the growing**
**season. All the determination coefficient $R^2$ and the Spearman rank correlation Rs are significant at 99%.**
**4.4 Relationship between backscattering coefficient and SSM**
Figure 12 displays the relationships between $\sigma^0$ and SSM using the entire database at 45.6° and 35.2° of incidence angles. H
is used as an indicator of vegetation growth. The correlation coefficient is computed separately for the entire database and
for data corresponding to H lower than a threshold value ($H_{tr}$) correspond to GLAI < 1.5. This value of GLAI correspond to
wheat not fully covering the soil (Ouaadi et al., 2020b). $H_{tr}$ is about 23.5 cm, 23.5 cm, 32.9 cm and 26 cm for F1 and F2
during 2017-2018, for F2 during 2016-2017 and for F3. Overall, $\sigma^0_{VV}$ is obviously better correlated to SSM than $\sigma^0_{VH}$ in line
with the results of numerous studies (Holah et al., 2005; Li et al., 2014; Ulaby and Batlivala, 1976). Likewise, metrics at
35.2° are better than those obtained at 45.6°. This is expected as the contribution of vegetation is dominant at higher
incidence angles and at VH polarization. The relationships are scattered when using data from the whole season. This is
attributed to the presence of vegetation and mainly to the attenuation of the soil signal backscattered by the wheat. The
sensitivity of $\sigma^0$ to SSM decreases progressively during the growing season as shown by the decreasing slope of the
relationships with the vegetation development. By considering the early season data only, when the soil is not yet covered by
vegetation, a better fitting is obtained between σ⁰ and SSM. Indeed, the correlation coefficient using data with $H < H_{tr}$ is
improved whatever the polarization and the incidence angle. Obviously, the highest correlation is obtained at VV
polarization and 35.2° of incidence angle (R = 0.73) and to a lesser extent at VV at 45.6° and VH at 35.2° with R ≥ 0.66.

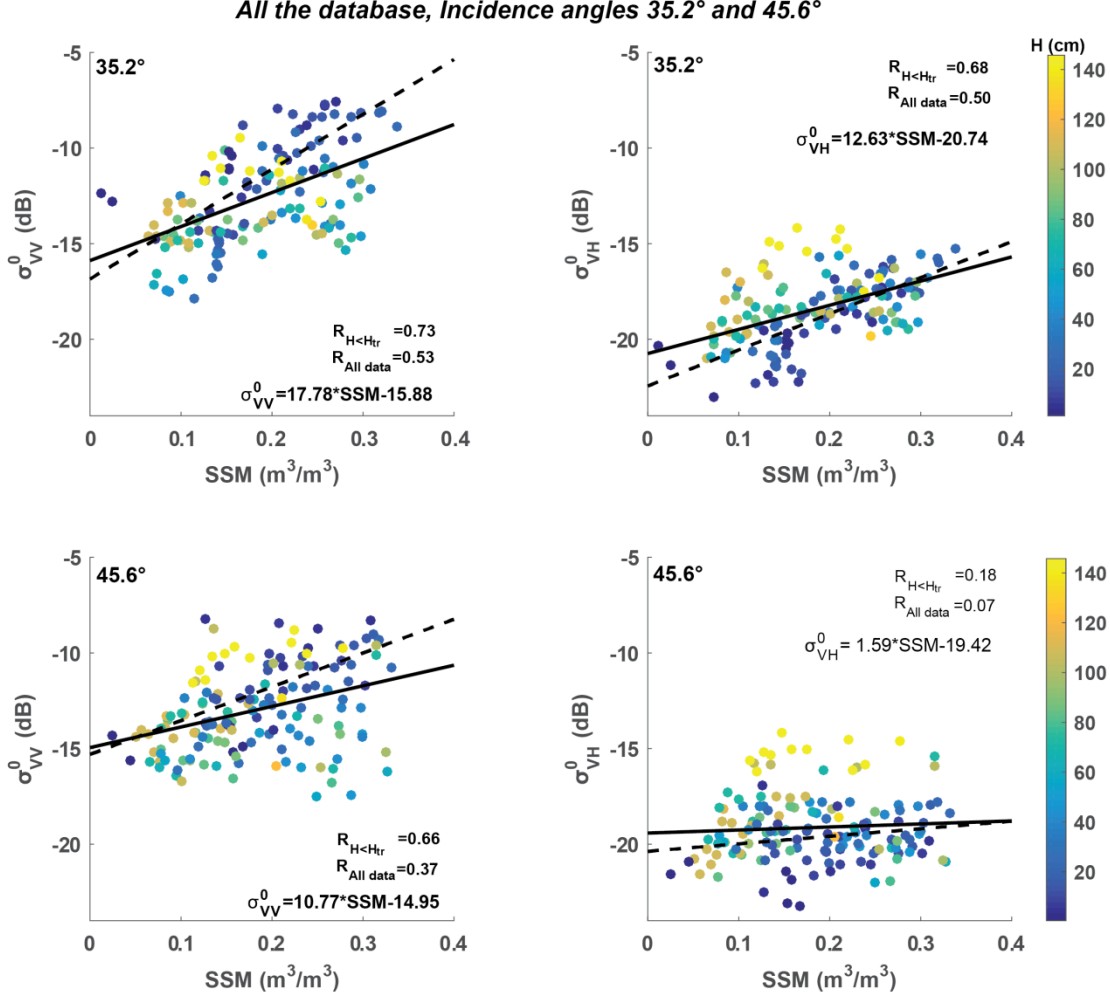

**Figure 12. Scatter plots of the relationships between $\sigma^0_{VV}$ and $\sigma^0_{VH}$ and SSM at 45.6° and 35.2 2° angles of incidence. The entire**
**database is used from the three fields (F1, F2 and F3). H is used to monitor the evolution during the growing season. The**
**significant correlation coefficients are in bold. The solid and the dashed lines correspond to all database and data with GLAI < 1.5,**
**respectively.**

## 5 Conclusion

This paper presents a 3-year database of C-band radar data and all necessary ancillary ground measurements to improve our understanding of the radar signal and to develop inversion methods for land surface parameters retrieval. The data are collected from three heavily monitored wheat fields under semi-arid conditions in the center of Morocco. The database offers a complete set of data for radar applications on wheat monitoring. The measured parameters include fresh and dry above ground biomass, canopy height, leaf area index, cover fraction, surface soil moisture, root zone soil moisture and surface roughness, in addition to the normalized difference vegetation index and SAR data (the backscattering coefficient and the interferometric coherence). The irrigation and meteorological data are also provided. This database opens the opportunity to use remote sensing together with measured parameters to understand and investigate the behavior of wheat crops and thereafter for vegetation parameters and soil moisture retrieval. The database analysis presented in this paper demonstrates the potentialities of SAR data for wheat monitoring by addressing the well-known sensitivity of SAR to surface soil moisture and vegetation variables. The obtained relationships between SAR measurements including backscattering coefficient, polarization ratio and interferometric coherence can be used for the application of several backscattering models, the retrieval of biophysical variables and for yield prediction in crop models. They can also be useful for land surface models relying on accurate estimation of vegetation height such as the energy balance models (i.e. TSEB -Two Source Energy Balance- Norman et al., 1995). The dataset illustrates also the complex signal acquired by C-band radar over wheat crops that is not yet fully understood as it mix the responses from highly dynamic contributions of soil and vegetation elements. The unique dataset provided in this paper should contribute through future studies to improve our understanding of the response of C-band radar observations over annual crops.

## 6 Database availability

This database is archived in DtaSuds repository of the French National Research Institute for Sustainable Development (IRD). The database is accessible free of charge with "CC-BY" licence at https://doi.org/10.23708/8D6WQC (Ouaadi et al., 2020a). It can be downloaded as xlsx files accompanied by a variable dictionary containing the variable names and units. The files are also accompanied by a metadata including a description of the database, time coverage, keywords and other general information.

**Appendix A: Complimentary figures**

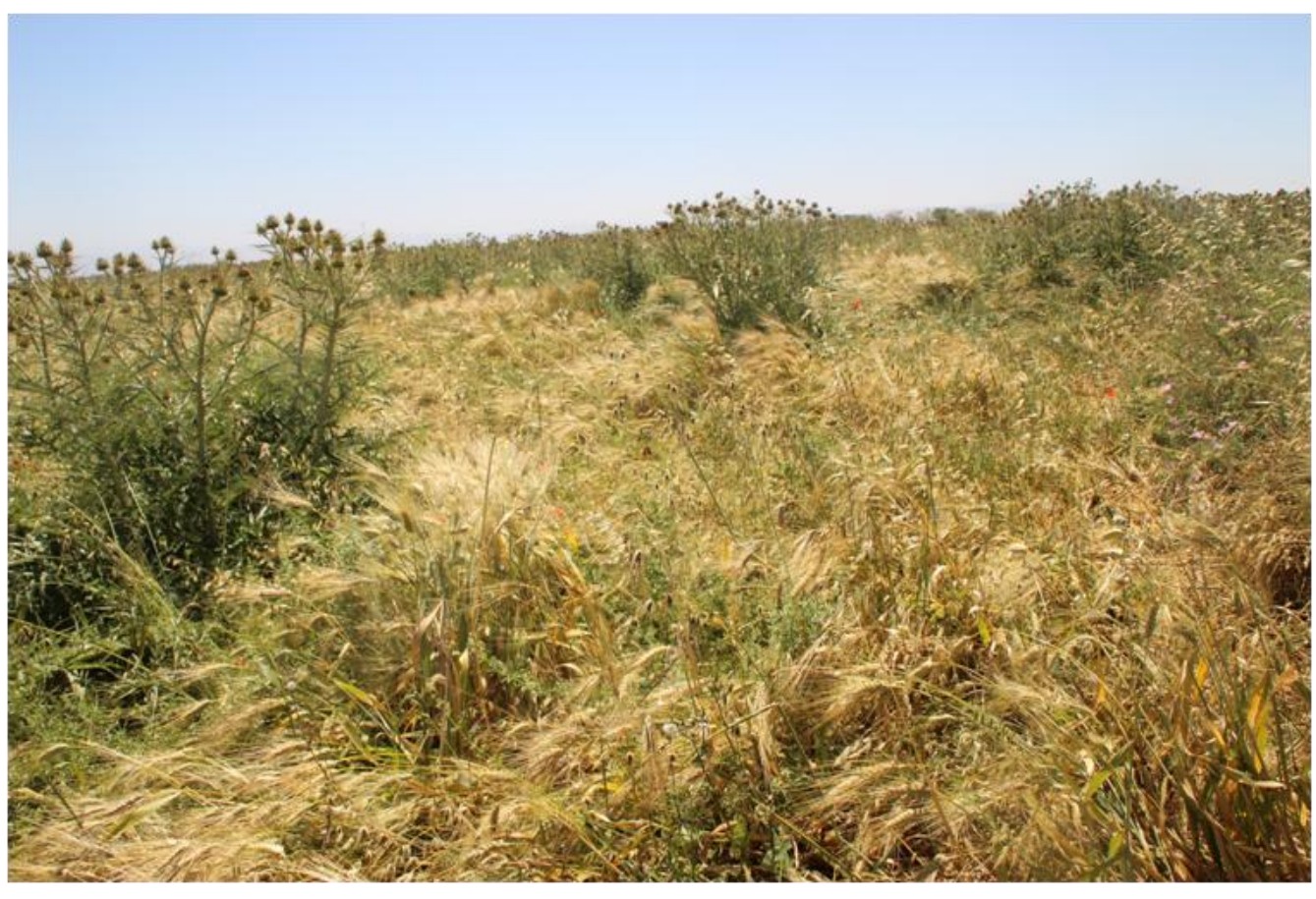


**Figure A1. Picture taken over F2 during 2017-2018 growing season (14/05/2018) illustrates the specific growing conditions**
**(adventices and stems laid down by wind).**

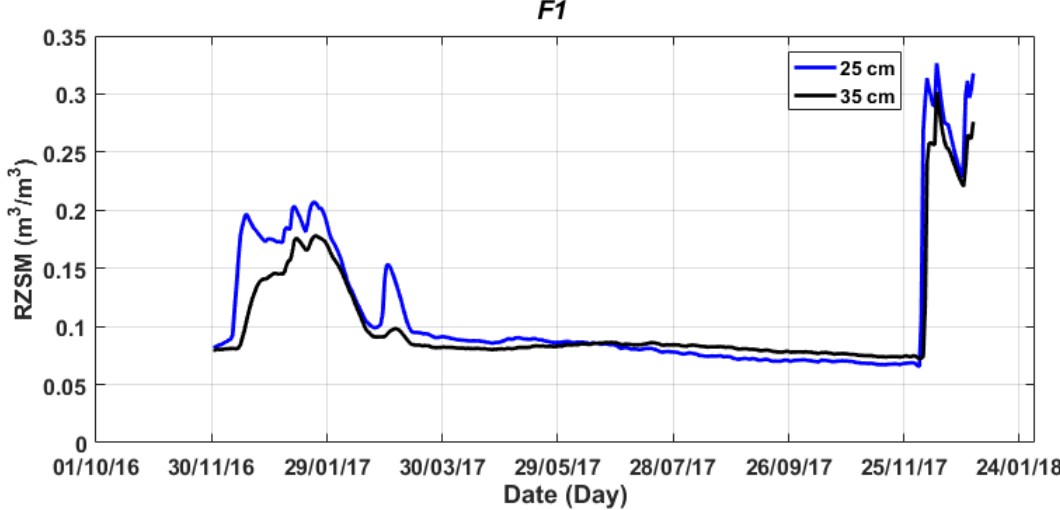


**Figure A2. Time series of root zone soil moisture (RZSM) at 25 and 35 cm of depth measured over F1 from December 01, 2016 to**
**December 31, 2017.**

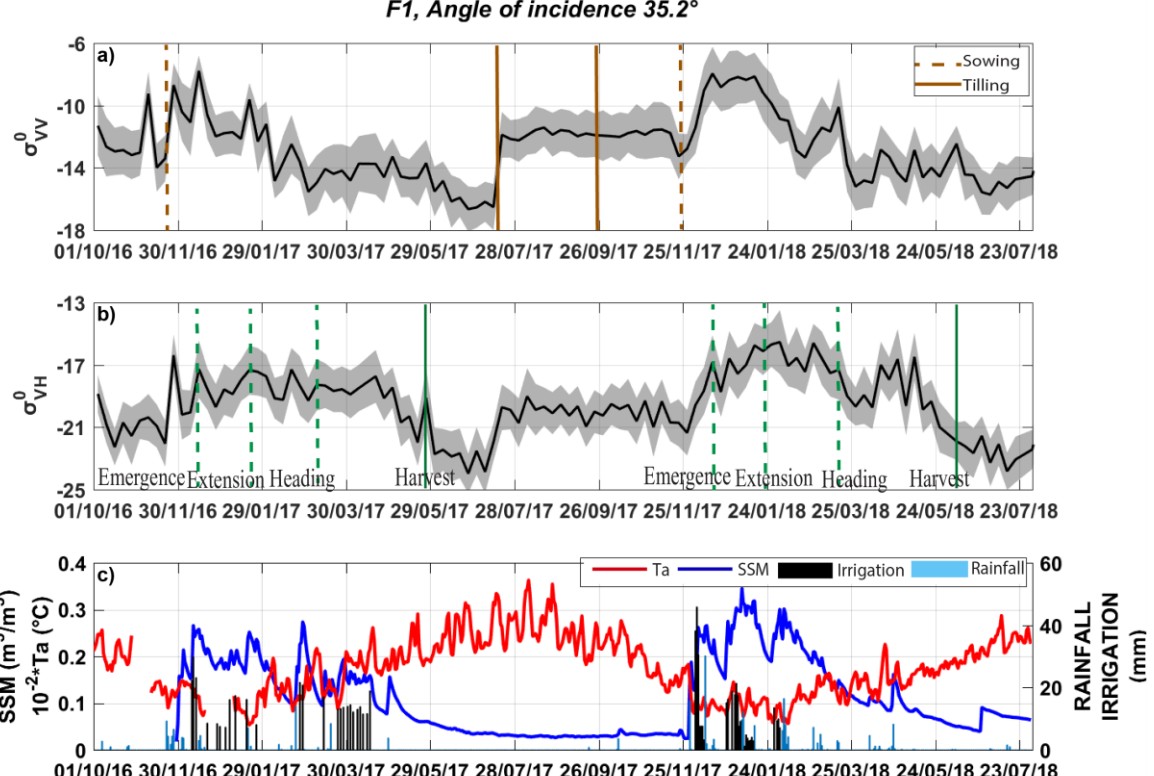

**Figure A3. Time series of the backscattering coefficient at VV (a) and VH (b) polarizations on F1 at 35.2° of incidence angle during**
**the period from October 01, 2016 to July 31, 2018. The tilling works and phenological stages of wheat are superimposed on**
**subplots (a) and (b), respectively. The air temperature, Surface soil moisture (SSM), irrigation and rainfall are displayed on**
**subplot (c).**

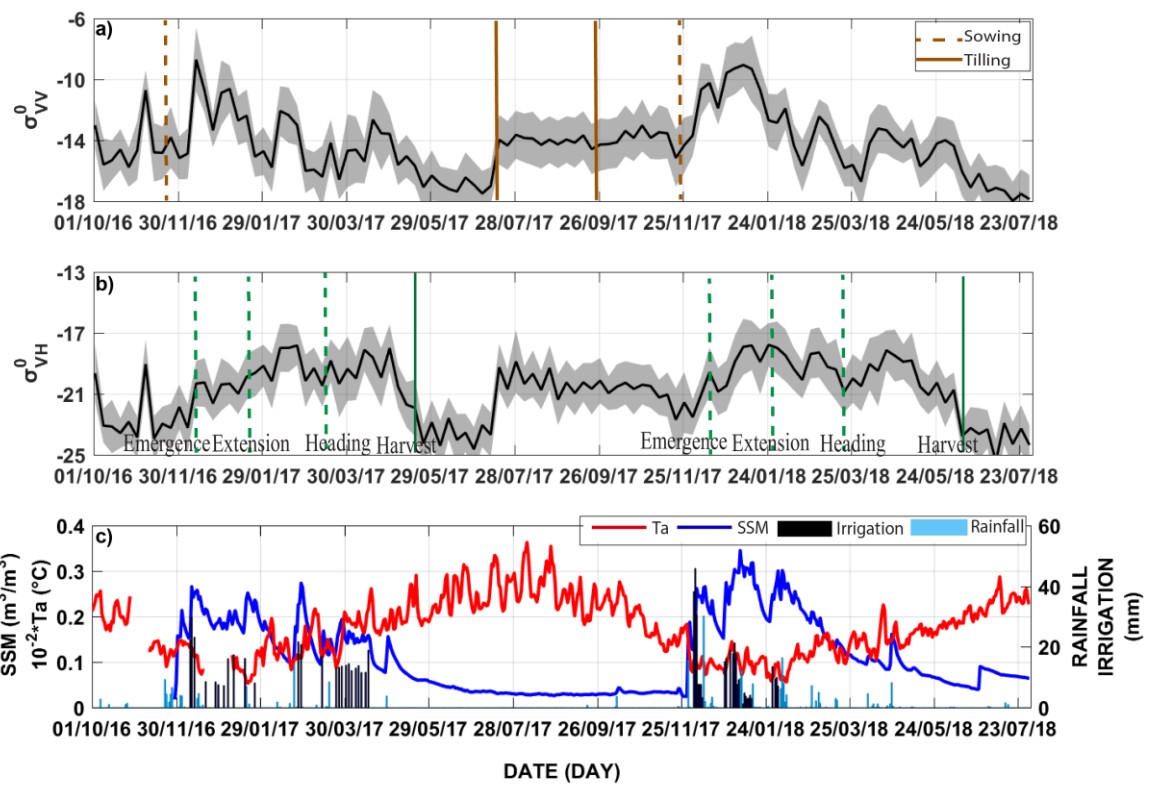

**Figure A4. Time series of the backscattering coefficient at VV (a) and VH (b) polarizations on F1 at 45.6° of incidence angle during**
**the period from October 01, 2016 to July 31, 2018. The tilling works and phenological stages of wheat are superimposed on**
**subplots (a) and (b), respectively. The air temperature, Surface soil moisture (SSM), irrigation and rainfall are displayed on**
**subplot (c).**

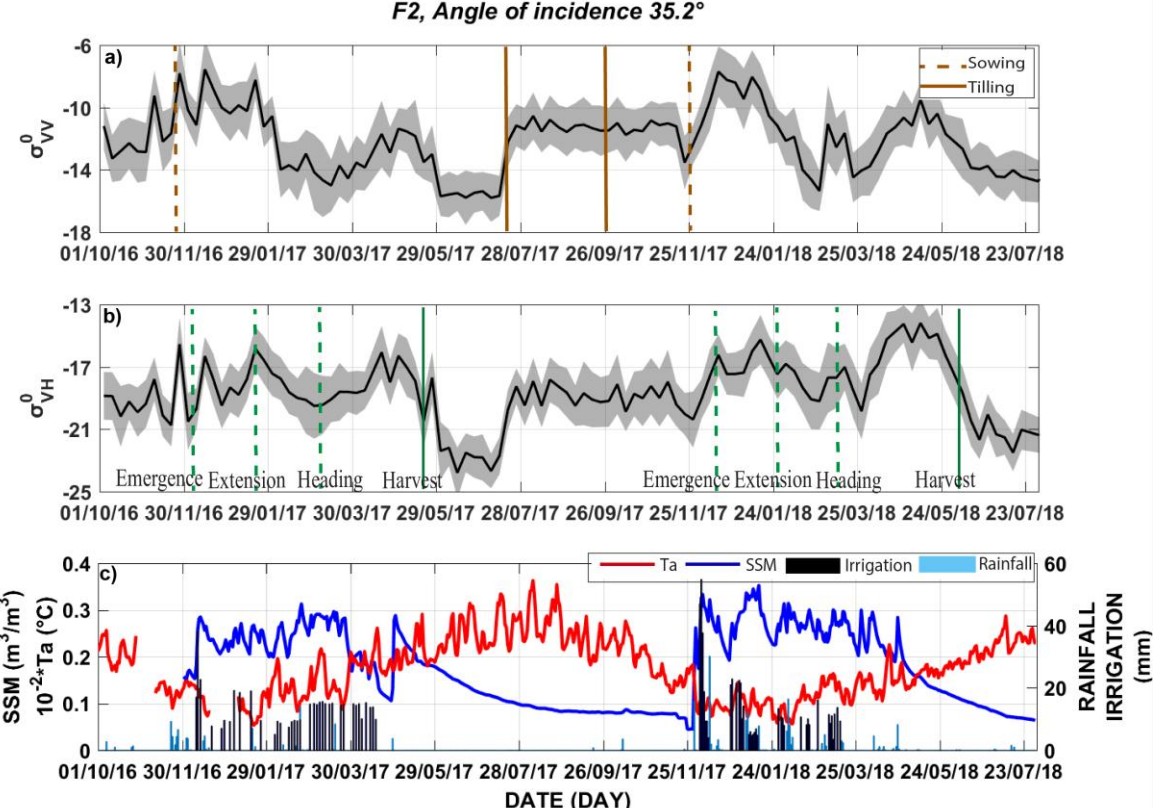

**Figure A5.** Time series of the backscattering coefficient at VV (a) and VH (b) polarizations on F2 at 35.2° of incidence angle during
the period from October 01, 2016 to July 31, 2018. The tilling works and phenological stages of wheat are superimposed on
subplots (a) and (b), respectively. The air temperature, Surface soil moisture (SSM), irrigation and rainfall are displayed on
subplot (c).

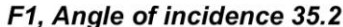

**F1, Angle of incidence 35.2°**

**Figure A6. Time series of the interferometric coherence at VV and VH polarizations (a) and the polarization ratio (b) on F1 at**
**35.2° of incidence angle during the period from October 01, 2016 to July 31, 2018. The tilling works and phenological stages of**
**wheat are superimposed on subplots (a) and (b), respectively. NDVI and measured GLAI are displayed in subplot (c). Measured**
**H, FAGB, VWC and AGB are plotted in subplot (d). Time series are presented by mean values (solid lines) and standard**
**deviations (filled fields surrounding the solid lines).**

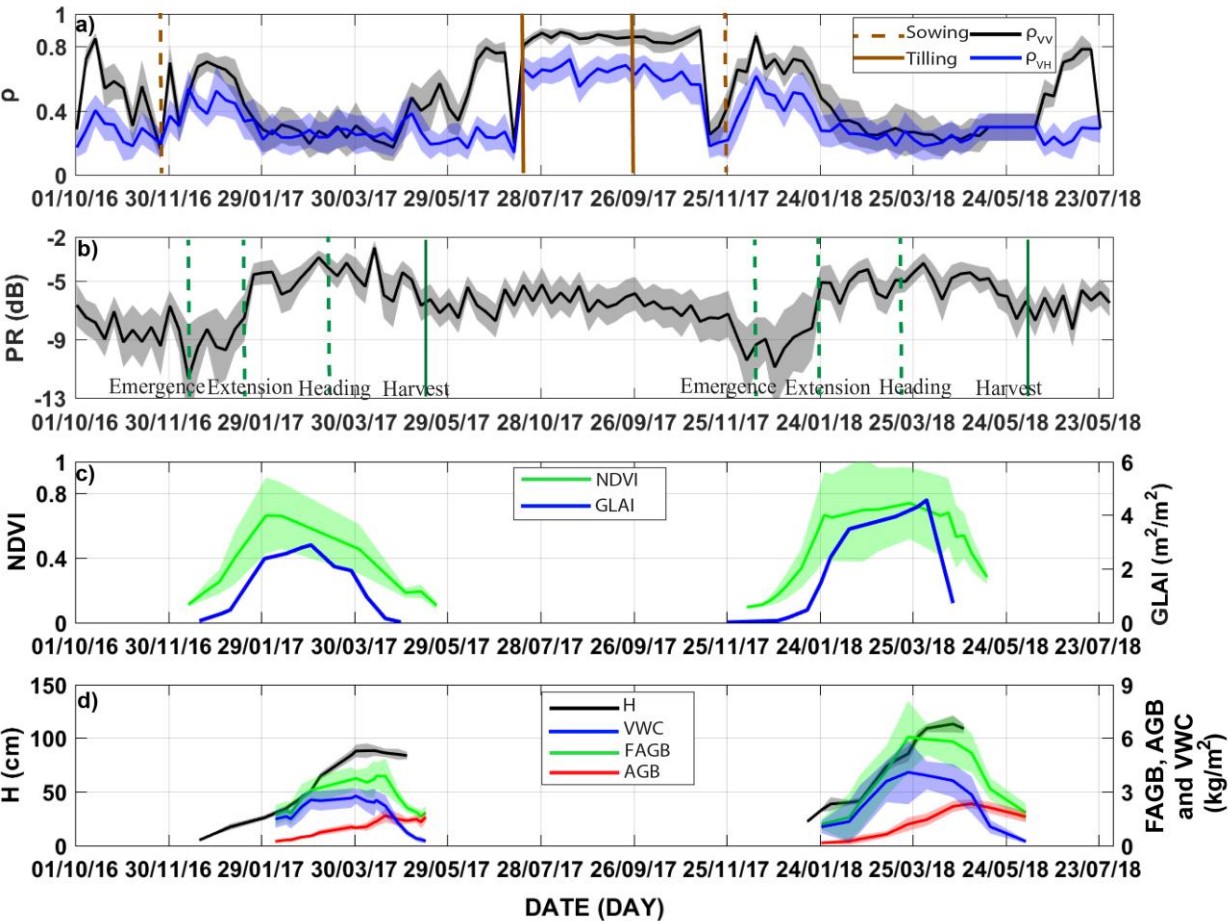

**Figure A7. Time series of the interferometric coherence at VV and VH polarizations (a) and the polarization ratio (b) on F1 at**
**45.6° of incidence angle during the period from October 01, 2016 to July 31, 2018. The tilling works and pheneological stages of**
**wheat are superimposed on subplots (a) and (b), respectively. NDVI and measured GLAI are displayed in subplot (c). Measured**
**H, FAGB, VWC and AGB are plotted in subplot (d). Time series are presented by mean values (solid lines) and standard**
**deviations (filled fields surrounding the solid lines).**

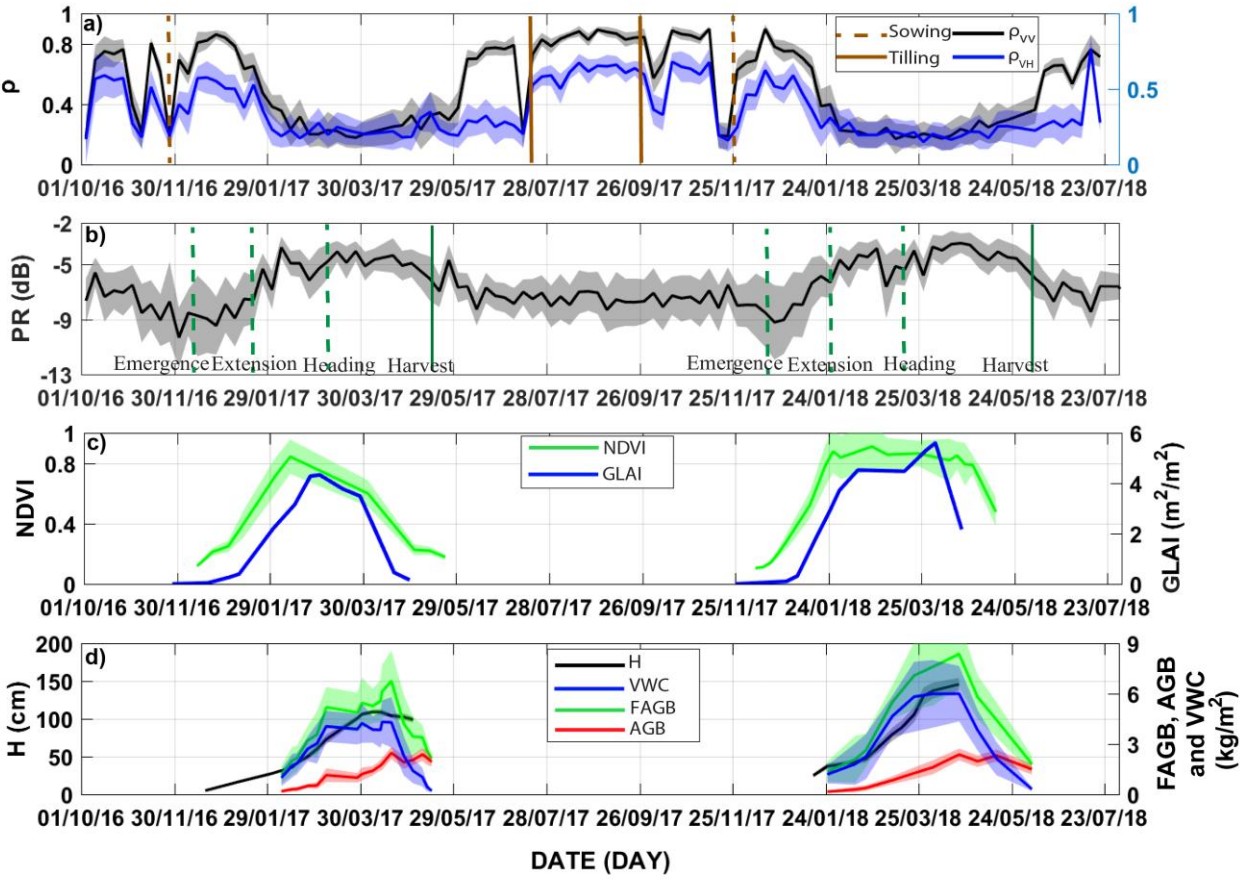

**Figure A8. Time series of the interferometric coherence at VV and VH polarizations (a) and the polarization ratio (b) on F2 at**
**35.2° of incidence angle during the period from October 01, 2016 to July 31, 2018. The tilling works and phenological stages of**
**wheat are superimposed on subplots (a) and (b), respectively. NDVI and measured GLAI are displayed in subplot (c). Measured**
**H, FAGB, VWC and AGB are plotted in subplot (d). Time series are presented by mean values (solid lines) and standard**
**deviations (filled fields surrounding the solid lines).**

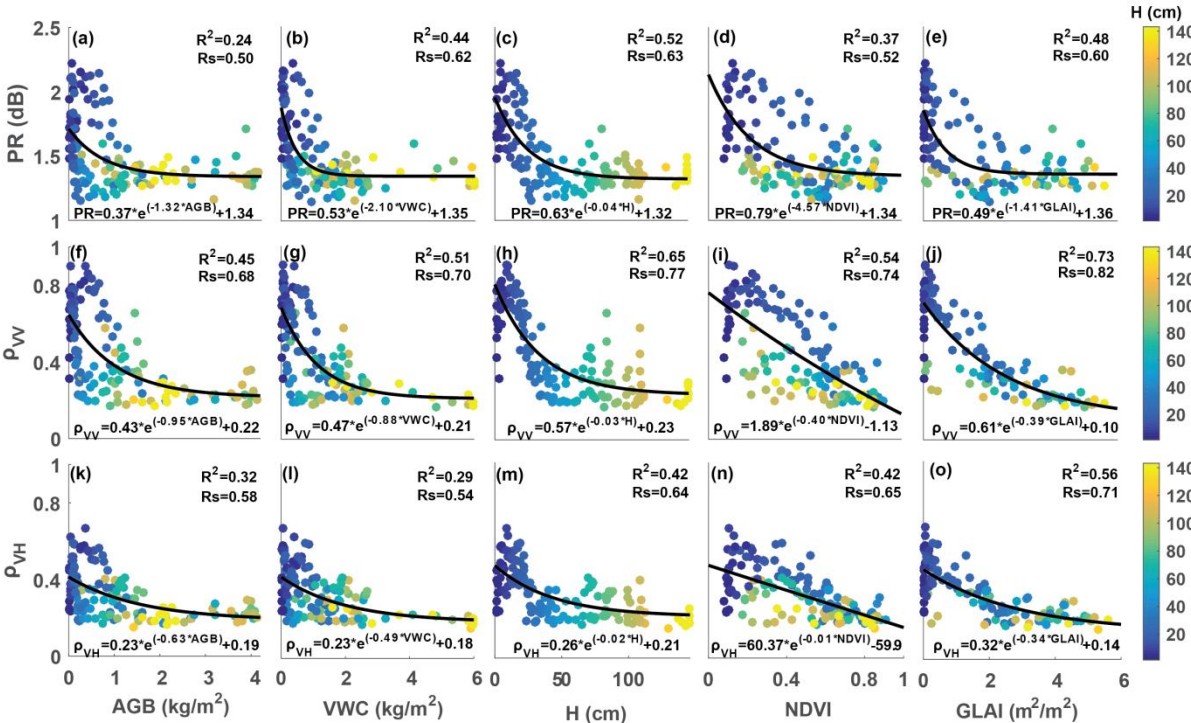


**Figure A9. Scatterplots of the relationships between PR, $\rho_{VV}$ and $\rho_{VH}$ and AGB, VWC, H, NDVI and GLAI at 45.6° angle of**
**incidence. The entire database is used from the three fields (F1, F2 and F3). H is used to monitor the evolution during the growing**
**season. All the determination coefficient $R^2$ and the Spearman rank correlation Rs are significant at 99%.**

**Table A1. Details of the field campaigns during the 2016-2017, 2017-2018 and 2018-2019 agricultural seasons.**

**2016-2017 saison**

| Date | $h_{rms}$ & L | H | Fc | LAI | AGB | FAGB | VWC |
|---|---|---|---|---|---|---|---|
| 29/11/2016 | x | | | | | | |
| 30/11/2016 | x | | | | | | |
| 09/12/2016 | x | | | | | | |
| 20/12/2016 | x | x | x | x | | | |
| 03/01/2017 | x | | x | x | | | |
| 09/01/2017 | x | x | x | x | | | |
| 31/01/2017 | x | x | x | x | | | |
| 07/02/2017 | | | | | x | x | x |
| 14/02/2017 | x | x | x | x | x | x | x |
| 17/02/2017 | | x | | x | x | x | x |
| 24/02/2017 | | x | x | x | x | x | x |
| 02/03/2017 | | x | x | x | x | x | x |
| 08/03/2017 | | x | | x | x | x | x |
| 17/03/2017 | | | x | x | | | |
| 28/03/2017 | | x | x | x | x | x | x |
| 31/03/2017 | | x | | x | x | x | x |
| 07/04/2017 | | x | x | x | x | x | x |
| 12/04/2017 | | x | | x | x | x | x |
| 13/04/2017 | | | | | x | x | x |
| 19/04/2017 | | x | x | x | x | x | x |
| 27/04/2017 | | x | | | x | x | x |
| 29/04/2017 | | | x | x | | | |
| 03/05/2017 | | x | | | x | x | x |
| 09/05/2017 | | | | | x | x | x |
| 12/05/2017 | | | | | x | x | x |
| 15/05/2017 | | | | | x | x | x |

**2017-2018 saison**

| Date | $h_{rms}$ & L | H | Fc | LAI | AGB | FAGB | VWC |
|---|---|---|---|---|---|---|---|
| 21/12/2017 | x | | | | | | |
| 28/12/2017 | x | | x | x | | | |
| 04/01/2018 | x | | x | x | | | |
| 16/01/2018 | x | x | x | x | | | |
| 25/01/2018 | x | x | x | x | x | x | x |
| 31/01/2018 | x | x | x | x | | | |
| 12/02/2018 | | x | x | x | x | x | x |
| 19/02/2018 | x | | | | | | |
| 08/03/2018 | x | | | | x | x | x |
| 14/03/2018 | x | x | x | x | x | x | x |
| 22/03/2018 | x | | | | x | x | x |
| 28/03/2018 | x | x | x | x | | | |
| 03/04/2018 | x | x | x | x | x | x | x |
| 20/04/2018 | x | x | x | x | x | x | x |
| 27/04/2018 | x | | | | | | |
| 02/05/2018 | | | | | x | x | x |
| 14/05/2018 | | | | | x | x | x |
| 06/06/2018 | | | | | x | x | x |

**2018-2019 saison**

| Date | $h_{rms}$ & L | H | Fc | LAI | AGB | FAGB | VWC |
|---|---|---|---|---|---|---|---|
| 29/11/2018 | x | | x | x | | | |
| 07/12/2018 | | | x | x | | | |
| 12/12/2018 | x | | | | | | |
| 18/12/2018 | | | x | x | | | |
| 04/01/2019 | x | | x | x | | | |
| 15/01/2019 | x | x | x | x | x | x | x |
| 01/02/2019 | | x | x | x | x | x | x |
| 13/02/2019 | | x | x | x | x | x | x |
| 04/03/2019 | | x | x | x | x | x | x |
| 13/03/2019 | | x | x | x | x | x | x |
| 26/03/2019 | | x | x | x | | | |
| 03/04/2019 | | x | | | | | |
| 09/04/2019 | | x | | | x | x | x |
| 16/04/2019 | | x | | | | | |
| 20/04/2019 | | x | | | x | x | x |
| 27/04/2019 | | x | | | x | x | x |


**Author contribution.** NO, LJ, JE and SK designed the experiments. NO, MK, JE, AC and AB carried the experiements out.
NO processed the Sentinel-1 products. NO, LJ, JE and SK analyzed the data. NO wrote the original draft and all the co-
authors contribute in the review and editing of the manuscript.
**Competing interests.** The authors declare that they have no conflict of interest.
**Acknowledgement.** The database is collected within the framework of the International Joint Laboratory TREMA
(https://www.lmi-trema.ma/). Omar Rafi, the owner of the private farm in which the three fields are located is
acknowledged. We would like to thank the projects: Rise-H2020-ACCWA (grant agreement no: 823965), ERANETMED03-
62 CHAAMS, PHC TBK/18/61 and MISTRALS/SICMED program. We thank also the Moroccan CNRST for awarding a
PhD scholarship to Nadia Ouaadi. Finally, ESA and Theia are acknowledged for providing free products of Sentinel-1 and
Sentinel-2 (corrected from atmospheric effects), respectively.

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
