# Peer review of "C-band radar data and in situ measurements for the monitoring of wheat crops in a semi-arid area (center of Morocco)"

_Earth System Science Data, 2020_

## Author Response (AR1)

**Response to reviewer 1**

We would like to thank the reviewer for his meaningful comment that we have taken into account. See our point-by-point response below.

l. 167: this is called the « auto-correlation function »

Right. The name was changed to "auto-correlation function" in the new version of the manuscript. See lines 171-172 in the marked-up version of the manuscript:

"hrms and L are computed from the **auto-correlation** function and then the average per direction, per field and per date is computed."

l. 182: replace « backscatter » by « radar backscattering response »

OK. Done. See lines 190-191 in the marked-up version of the manuscript:

"Biomass and water content are two biophysical parameters of crucial importance in different agricultural applications including particularly plant stress monitoring, radar backscattering response, crop yield and evapotranspiration modeling"

§ 3.1.4: Add references detailing the used GLAI estimation method.

OK. Two references were added as requested by the reviewer: Duchemin et al., 2006 and Khabba et al., 2009. See lines 202-204 in the marked-up version of the manuscript:

"H is simply measured using a measuring tape while GLAI and Fc are computed by processing hemispherical photos (Fig. 6b) using MATLAB software **following the method described in Duchemin et al. (2006) and Khabba et al. (2009)**."

l. 241-256: Add a reference advocating this data processing chain

OK. The reference to Frison and Lardeux, 2018 was added in the new version of the manuscript. The same processing is also used by Bousbih et al. (2017). See lines 257-258 in the marked-up version of the manuscript:

"The processing procedure consists of three steps **(Frison and Lardeux, 2018)** "

l. 259-260: Give the number of pixels involved in the average process, the corresponding ELN estimation, and the corresponding error on the estimated backscattering coefficient.

Right. In response to the reviewer comment, the number of pixels and the error on the backscattering coefficient is now added in the new version of the manuscript. By contrast, we didn't understand what ELN means. See lines 27-274 in the marked-up version of the manuscript:

"The presented database is generated using a simple average per field **of 120, 121 and 1100 pixels for F1, F2 and F3, respectively with a mean standard deviation of around 1.55 dB.**"

l. 261-293: - Add a reference advocating this data processing chain

OK. The reference to Veci, 2015 is added in the new version of the manuscript. See line 294-295 in the marked-up version of the manuscript:

"Sentinel application platform SNAP is used to compute the interferometric coherence from S1-SLC products in five steps **(Veci, 2015)**"

- Specify the spatial neigbourhood dimension in range and azimuth that is used.

OK. The spatial neighborhood dimension range*azimuth was specified at line 289. See lines 305-306 in the marked-up version of the manuscript:

"In order to get a square pixel of 13.95 m, **azimuth* range are fixed to 3*15** in the processing."

l. 298: Give details nabout the atmospheric corrections made for the Sentinel-2 level-2A products (the atmospheric radiative transfer model as well as the atmospheric gazeous and aerosols concentrations that are used)

OK. The correction chain used for S2 correction is named MAYA and has been developed by Hagolle et al, 2015. The atmospheric corrections are performed in three steps:

1-The satellite top-of-atmosphere (TOA) reflectances are corrected from the absorption by the atmospheric gas molecules usingthe absorption part of the Simplified Model for Atmospheric Correction (SMAC) method by Rahman et al., 1994. The concentrations of the ozone, the oxygen and the water vapor are obtained from satellite data (ozone) and meteorological data (water vapor, pressure).

2- The detection of the clouds (and cloud's shadows) is based on the multi-temporal cloud detection method proposed by Hagolle et al., 2010 .

3-The estimation of the aerosol optical thickness (AOT) relies on a hybrid method merging the criteria of a multi-spectral method with the multi-temporal technique developed initially for the VENμS satellite mission by Hagolle et al., 2010. The AOT is used along with the surface altitude, the viewing geometry and the wavelength in the parameterization of look-up tables for the conversion of TOA reflectances already corrected in step "1" into surface reflectances. The look-up tables are provided by the successive orders of scattering code (Lenobel et al., 2007) used in the modeling of molecular and aerosol scattering effects. A different look-up table is computed for each aerosol model.

In response to the reviewer comment, the processing chain is now described in the new version of the manuscript (see lines 316-330 in the marked-up version of the manuscript):

**"The atmospheric corrections are performed in three steps:**

1. **The satellite top-of-atmosphere (TOA) reflectances are corrected from the absorption by the atmospheric gas molecules using the absorption part of the Simplified Model for Atmospheric Correction (SMAC) method by Rahman et al., 1994. The concentrations of the ozone, the oxygen and the water vapor are obtained from satellite data (ozone) and meteorological data (water vapor, pressure).**

2. **The detection of the clouds (and cloud's shadows) is based on the multi-temporal cloud detection method proposed by Hagolle et al., 2010 .**

**3. The estimation of the aerosol optical thickness (AOT) relies on a hybrid method merging the criteria of a multi-spectral method with the multi-temporal technique developed initially for the VENµS satellite mission by Hagolle et al., 2010. The AOT is used along with the surface altitude, the viewing geometry and the wavelength in the parameterization of look-up tables for the conversion of TOA reflectances already corrected in step "1" into surface reflectances. The look-up tables are provided by the successive orders of scattering code (Lenobel et al., 2007) used in the modeling of molecular and aerosol scattering effects. A different look-up table is computed for each aerosol model."**

l. 378: Precise that Polarization Ratio consist in the ratio between Sigma0_VH / Sigma0_VV

Agree. This is now specified in the new version of the manuscript. See line 416 in the marked-up version of the manuscript:

"b) polarization ratio (PR = $\sigma^0_{VH}/\sigma^0_{VV}$)"

Fig. 9, 10, A3-A8: Are these temporal profiles similar than those already published in litterature? A comparison should be welcome.

Agree. Similar temporal profiles were partly already published. For the backscattering coefficient, the temporal behavior of wheat over a growing season at C-band was found by several authors to be characterized by four stages; i) the signal is first governed by soil moisture dynamic during the first growing stage, ii) The backscattering coefficient decreases in a second step under the effect of canopy attenuation until the heading stage when it reaches a minimum value and iii) it increases again gradually in response to the development of the head creating a thin very wet layer at the top of the canopy favoring volume backscattering; iv) Finally, backscattering decreases during senescence in response to the soil and the vegetation drying. The first part differs from one study to another given the differences in soil hydric condition and surface roughness of the sites. After this period, the behavior of the signal is overall similar to the profiles obtained by Cookmartin et al. (2000), El Hajj et al. (2019), Nasrallah et al. (2019) and Veloso et al. (2017). With the development of vegetation, the decrease caused by the canopy attenuation has been observed by several authors before, as indicated in the manuscript (Cookmartin et al., 2000; Mattia et al., 2003; Picard et al., 2003; Wang et al., 2018). After heading, the increase of the backscattering coefficient at C-band was reported first by Ulaby and Batlivala (1976). Ulaby et al. (1986) suggested that an additional term needs to be added to the traditional three terms model (volume scattering from vegetation, soil attenuated and interaction soil-vegetation) to properly represent wheat backscattering after heading. The increase after heading has then be observed and attributed to the appearance of the ears followed by the grain by numerous authors (Brown et al., 2003; El Hajj et al., 2019; Mattia et al., 2003; Patel et al., 2006; Veloso et al., 2017; Ouaadi et al., 2020). In response to the reviewer comment, the comparison to literature has been strengthened in the new version of the manuscript.

For the interferometric coherence, only a few time series have been presented and analyzed on wheat crops to our knowledge. In line with the time series illustrated in this work (Figures 10 and A8), Santoro et al., 2010 demonstrates using the ERS−Envisat Tandem mission that

coherence measurements of vegetated fields are always below the level of bare soils coherence. During the period of vegetation, the observed degradation/decrease of coherence with wheat development have been illustrated by Blaes and Defourny (2003) and Engdahl et al. (2001) even that the number of data was limited (less than six data along the season). In response to the reviewer comment, this is now specified in the new version of the manuscript.

See lines 378-395 in the marked-up version of the manuscript:

"**Although it is agreed that the signal during this period is governed by the dynamics of soil moisture, its behavior differs from one site to another giving the difference in soil hydric conditions and surface roughness. After this period, the signal behavior is similar to the profiles obtained by Cookmartin et al. (2000), El Hajj et al. (2019), Nasrallah et al. (2019) and Veloso et al. (2017).** It decreases gradually from the early tillering until the heading stage (around March 13) by about 10 dB on F2 and 5 dB on F1 because of the attenuation by the canopy during the development of the stems (extension stage) **(Cookmartin et al., 2000; Mattia et al., 2003; Picard et al., 2003; Wang et al., 2018)**. Obviously, the attenuation is more important at VV polarization because of the vertical structure of wheat (stems) in line with the results of (Fontanelli et al., 2013; Picard et al., 2003; Wang et al., 2018). The response of $\sigma_{VH}^{0}$ to SSM variation and canopy attenuation is lower than for $\sigma_{VV}^{0}$. After the heading stage, the signal starts to increase again. This is clearer on F2 than F1 and at 45.6° than at 35.2°. The heading stage is the phenological stage of wheat when the spike or head starts emerging out from the leaf sheath. This change of the structure of the canopy shield the stems for the radar signal through the appearance of a thick, wet top layer composed of the heads. The C-band wavelength penetrates this layer only, resulting in increased volume scattering, while attenuation becomes low. This effect is stronger for F2 than for F1, at VH than at VV and at 45.6° than at 35.2°. **This increase was first reported by Ulaby and Batlivala (1976). Subsequently, Ulaby et al. (1986) suggested that an additional term must be added to the traditional three-term model (vegetation volume diffusion, soil attenuation, and soil-vegetation interaction) to properly represent wheat backscattering after heading. Later on, similar behaviour has been observed and attributed to the appearance of the heads followed by the grain by numerous authors (Brown et al., 2003; El Hajj et al., 2019; Mattia et al., 2003; Patel et al., 2006; Veloso et al., 2017).**"

And lines 426-430:

"After sowing**, the evolution is similar to the profiles obtained by Blaes and Defourny (2003) and Engdahl et al. (2001).** The interferometric coherence increases from 0.15 to 0.7 and then starts to decrease slightly from the emergence of wheat, becoming almost constant after stem extension with values < 0.3 corresponding to the noise level. **Indeed, using the ERS–Envisat Tandem mission, Santoro et al., (2010) demonstrated that coherence measurements of vegetated fields are always below the level of bare soils coherence.**"

Agree. The unit is the same for VWC, FAGB and AGB ($kg/m^2$). The three variables were grouped under the nomination biomass on the y-axis for simplification in the previous version. In response to the reviewer's comment, the label of the y-axis was changed to "ABG, FABG and VWC (kg/m²)". The same was done for Fig. A6-A8.

[Figure]

**Figure 1. Time series of the interferometric coherence at VV and VH polarizations (a) and the polarization ratio (b) on F2 at 45.6° of incidence angle during the period from October 01, 2016 to July 31, 2018. The tilling works and phenological stages of wheat are superimposed on subplots (a) and (b), respectively. NDVI and measured GLAI are displayed in subplot (c). Measured H, FAGB, VWC and AGB are plotted in subplot (d). Time series are presented by mean values (solid lines) and standard deviations (filled fields surrounding the solid lines).**

[Figure]

**Figure 2.** Time series of the interferometric coherence at VV and VH polarizations (a) and the polarization ratio (b) on F1 at 45.6° of incidence angle during the period from October 01, 2016 to July 31, 2018. The tilling works and phenological stages of wheat are superimposed on subplots (a) and (b), respectively. NDVI and measured GLAI are displayed in subplot (c). Measured H, FAGB, VWC and AGB are plotted in subplot (d). Time series are presented by mean values (solid lines) and standard deviations (filled fields surrounding the solid lines).

[Figure]

**Figure 3.** Time series of the interferometric coherence at VV and VH polarizations (a) and the polarization ratio (b) on F1 at 35.2° of incidence angle during the period from October 01, 2016 to July 31, 2018. The tilling works and phenological stages of wheat are superimposed on subplots (a) and (b), respectively. NDVI and measured GLAI are displayed in subplot (c). Measured H, FAGB, VWC and AGB are plotted in subplot (d). Time series are presented by mean values (solid lines) and standard deviations (filled fields surrounding the solid lines).

[Figure]

**Figure 4.** Time series of the interferometric coherence at VV and VH polarizations (a) and the polarization ratio (b) on F2 at 35.2° of incidence angle during the period from October 01, 2016 to July 31, 2018. The tilling works and phenological stages of wheat are superimposed on subplots (a) and (b), respectively. NDVI and measured GLAI are displayed in subplot (c). Measured H, FAGB, VWC and AGB are plotted in subplot (d). Time series are presented by mean values (solid lines) and standard deviations (filled fields surrounding the solid lines).

4. The reference need to be modified carefully. For example, Line 551 with Uppercase article title;

Thank you. All the references were checked and modified when needed.

5. Line 146 I guess it is 0.018?

Yes, RMSE=0.018 m$^3$/m$^3$. Thank you, the comma is replaced by a dot in the new version of the manuscript (see line 150 in the marked-up version of the manuscript).

6. Add reference for the vegetation water content process.

OK. The reference to Gherboudj et al. (2011) is added in the new version of the manuscript ( lines 196-197 in the marked-up version of the manuscript):

"The vegetation water content (VWC) is thus computed as the difference between FAGB and AGB **(Gherboudj et al., 2011)."**

7. How do you consider the effect of precipitation on the surface roughness?

In addition to irrigation, rain is supposed to impact slightly the roughness in the beginning of the crop season (before the wheat covers the soil) as the rows are directly exposed to rainfall.

During this period, the roughness is measured every week/two weeks to take into account the effect of precipitation and irrigation. After this period, the roughness is assumed to be constant. Indeed, it has been shown in literature that after sowing (no soil works happened), roughness is only affected by very limited temporal variations (Bousbih et al., 2017) and it is generally kept constant during the crop season (El Hajj et al., 2016; Gherboudj et al., 2011; Gorrab et al., 2015; Ouaadi et al., 2020). In response to the reviewer comment, this is now clarified in the new version of the manuscript, lines 178-185 in the marked-up version of the manuscript:

"After sowing, **a slight change is observed at the start of the crop season (December 28, 2017, see Fig. 4). **At that time, the soil has just been prepared for sowing and **rows are directly exposed to rain. The fact that the rows are still visible in the field also explains the differences observed between both directions early in the season**. This anisotropy disappeared quickly with irrigation, rainfall and plant growth. **hrms and L are almost constant from early January onwards. Indeed, it has been shown that after sowing, roughness is affected by very limited temporal variations (Bousbih et al., 2017) as no soil works occur after sowing. It is usually kept constant during the crop season (El Hajj et al., 2016; Gherboudj et al., 2011; Gorrab et al., 2015; Ouaadi et al., 2020).**"

8. How to measure the surface roughness during the extension growth stage?

OK. With a pin profiler, the measurements of surface roughness when the canopy covers the soil are almost impossible explaining why the data base extends during the first stages of wheat growth. It is assumed to be constant after this time (see response to point 7).

9. Under what condition will you start irrigation?

OK. The irrigation process is driven by the farmer based on evapotranspiration demand computed with the FAO-56 simple approach (Allen et al., 1998; lines 206-207). The timing of irrigation is determined by the farmer according to the available workforce, the occurrence of rain... This is now clarified in the new version of the manuscript (lines 216-220 in the marked-up version of the manuscript):

"Irrigation quantities are determined by the farmer by estimating **the daily evapotranspiration under standard conditions (ETc) in the region computed using the FAO-56 model simple approach (Allen et al., 1998). The cumulative ETc for a given period (usually one week) is applied during one or more events per week depending on the farmer's constraints (e.g. availability of workforce) and on the weather conditions (e.g. occurrence of rain).**"

10. How many times of field observations do you made? and can you list the specific information of each filed campaign?

OK. The numbers of field's campaigns are 26, 18 and 16 campaigns during 2016-2017, 2017-2018 and 2018-2019 seasons, respectively. The table below summarizes the campaign details. In response to the reviewer suggestion, this table is added in the new version of the manuscript (lines 127-130 in the marked-up version of the manuscript):

[revised manuscript text omitted]